# Matrix mechanical plasticity regulates cancer cell migration through confining microenvironments

Katrina M. Wisdom [1], Kolade Adebowale [2], Julie Chang[3], Joanna Y. Lee [1], Sungmin Nam[1], Rajiv Desai[4], Ninna Struck Rossen[5], Marjan Rafat [5], Robert B. West[6], Louis Hodgson [7] & Ovijit Chaudhuri [1]

Studies of cancer cell migration have found two modes: one that is protease-independent, requiring micron-sized pores or channels for cells to squeeze through, and one that is protease-dependent, relevant for confining nanoporous matrices such as basement membranes (BMs). However, many extracellular matrices exhibit viscoelasticity and mechanical plasticity, irreversibly deforming in response to force, so that pore size may be malleable. Here we report the impact of matrix plasticity on migration. We develop nanoporous and BM ligand-presenting interpenetrating network (IPN) hydrogels in which plasticity could be modulated independent of stiffness. Strikingly, cells in high plasticity IPNs carry out protease-independent migration through the IPNs. Mechanistically, cells in high plasticity IPNs extend invadopodia protrusions to mechanically and plastically open up micron-sized channels and then migrate through them. These findings uncover a new mode of protease-independent migration, in which cells can migrate through confining matrix if it exhibits sufficient mechanical plasticity.

[1] Department of Mechanical Engineering, Stanford University, Stanford, CA 94305, USA. [2] Department of Chemical Engineering, Stanford University, Stanford, CA 94305, USA. [3] Department of Bioengineering, Stanford University, Stanford, CA 94305, USA. [4] School of Engineering and Applied Sciences, Harvard University, Cambridge, MA 02138, USA. [5] Department of Radiation Oncology, Stanford University, Stanford, CA 94305, USA. [6] Department of Clinical Pathology, Stanford University, Stanford, CA 94305, USA. [7] Department of Anatomy and Structural Biology, Gruss-Lipper Biophotonics Center, Albert Einstein College of Medicine, Bronx, NY 10461, USA. Correspondence and requests for materials should be addressed to O.C. (email: chaudhuri@stanford.edu)

Carcinoma progression and metastasis require that cancer cells traverse basement membranes (BMs): first through the BM separating epithelial and stromal tissue, and then across the BM lining blood vessels (Fig. 1a)[1,2]. Invadopodia are the actin-rich, invasive protrusions that enable cancer cells to invade the BM, and they are thought to do so by secreting proteases to degrade the BM[3,4]. Recent studies suggest that without matrix degradation, nanometer-scale pores of BM would physically limit invasion, as cells are unable to squeeze through elastic or rigid pores smaller than roughly 3–5 μm in diameter[5–11]. However, pore size may be malleable—particularly in tumor tissue. While it has been long appreciated that tumor tissue is up to an order of magnitude stiffer than normal tissue[12], noninvasive clinical imaging has also revealed breast tumor tissue to be more viscous, or liquid-like, than normal tissue[13]. The elevated viscosity of tumor tissue is thought to arise in part from abnormal tissue cross-linking that accompanies breast cancer progression[13,14]. Because matrix plasticity can be related to matrix viscosity, matrix architecture in the tumor microenvironment may also exhibit elevated mechanical plasticity, enabling cell-generated forces to induce permanent microstructural rearrangements in the matrix. This raises the possibility that cells can carry out invasion into, and migration through, confining matrices using cell-generated forces to dilate pores if those matrices are sufficiently plastic.

Here we assess the role of matrix plasticity in mediating invasion and migration of confined cancer cells. We develop IPN hydrogels, which are nanoporous, present BM ligands to cells, and enable mechanical plasticity to be modulated independent of stiffness. Cells in high plasticity (HP) IPNs carry out protease-independent migration through the IPNs, while cells in low plasticity (LP) IPNs mostly do not migrate. Cells in HP IPNs first utilize invadopodia protrusions to mechanically and plastically open up channels, and then generate protrusive forces at the leading edge to migrate through them. Together, these findings establish that cells can migrate through confining matrices such

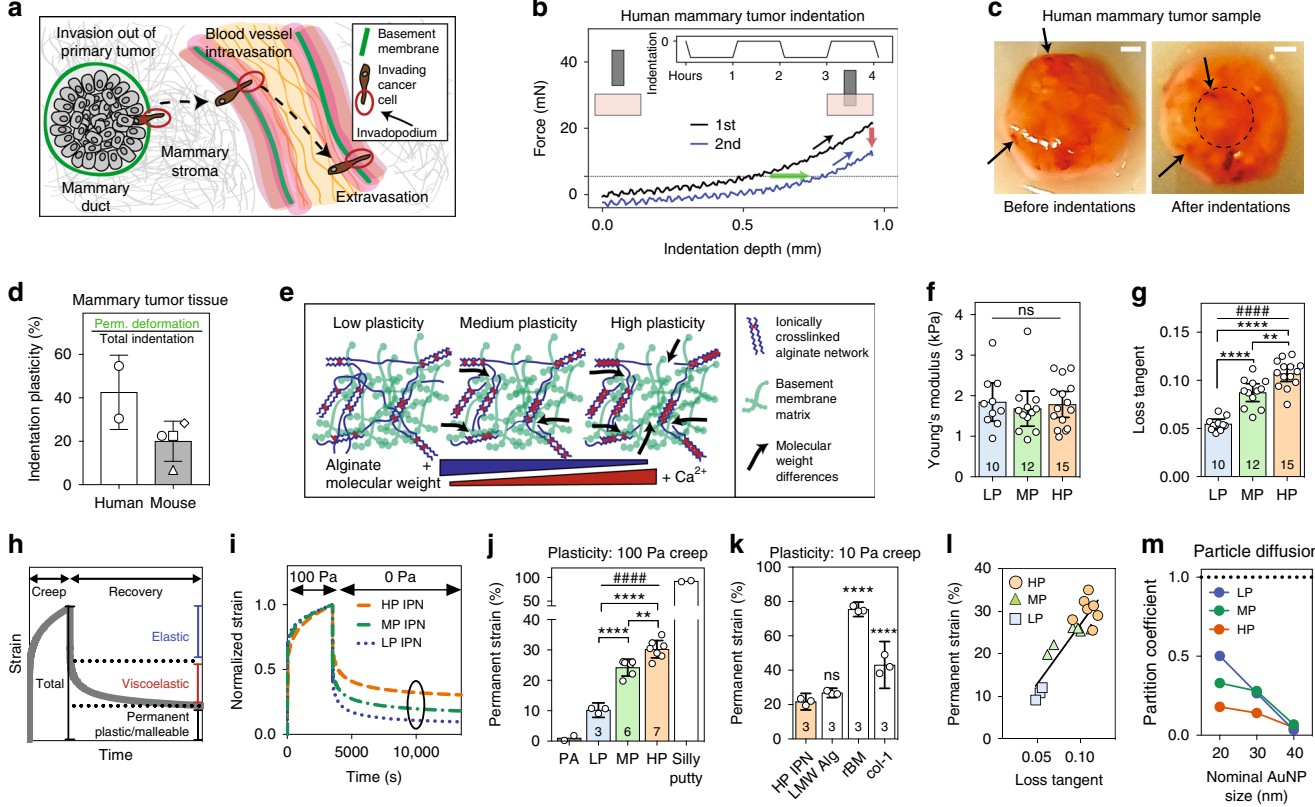

**Fig. 1** Mechanical plasticity of interpenetrating networks of alginate and reconstituted basement membrane matrix (IPNs) can be independently tuned. **a** Schematic depicting invasion of basement membranes (green) during invasion and metastasis. **b** Schematic depicting the indentation tests performed on human mammary tumor tissue, and the corresponding force vs. indentation depth curves (green arrow—permanently retained indentation; red arrow—drop in peak force during second indentation; dotted line—25% of initial peak force). Subplot shows indentation test profile. **c** Before and after images of an indented mammary tumor sample. Indentation region outlined by dotted circle, and discolored tissue regions indicated by black arrows. Scale bar is 1 mm. **d** Indentation plasticity measurements of human tumor (two specimens from a tumor sample) and mouse tumor specimens (one sample each from four separate mice). **e** Schematic of approach to tuning matrix plasticity in IPNs of alginate (blue) and reconstituted basement membrane (rBM) matrix (green). **f**, **g** Young's moduli (**f**) and loss tangent (**g**) of the different IPN formulations. The differences in loss tangent indicated are significantly different (**P < 0.01, ****P < 0.0001, ANOVA; ns not significant), as is the increasing loss tangent across this series of IPNs (####P < 0.0001, Spearman's rank correlation). **h** Schematic depicting the elastic, viscoelastic, and plastic (permanent) portions of a material response in a creep and recovery test. **i** Representative creep and recovery tests of IPNs. **j** Permanent strain of IPNs, polyacrylamide gels (PA), and silly putty from creep and recovery tests. Statistically significant differences are indicated (**P < 0.01, ****P < 0.0001, ANOVA), plasticity across the IPNs (####P < 0.0001, Spearman's rank correlation). **k** Permanent strain of HP IPN, alginate hydrogel, rBM matrix, and col-1 gels from creep–recovery tests. Statistically significant differences compared to HP IPN are indicated (****P < 0.0001, ANOVA). In **f–k**, bars indicate means and error bars indicate 95% confidence interval of the indicated biological replicates. **l** Permanent strain and loss tangent are correlated in the IPNs ($R^2 = 0.7953$). **m** Partition coefficients for PEGylated gold nanoparticles of the indicated size encapsulated in IPNs for 4 days

as BMs or dense stromal tissue independent of proteases, if the matrix exhibits mechanical plasticity.

## Results

**Mechanical plasticity in breast tumors**. First, we assessed mechanical plasticity in breast tumors. Indentation testing of human breast tumor tissue revealed that these tissues exhibit substantial mechanical plasticity, as they permanently sustained substantial deformations following indentation and exhibited a loss in the peak force sustained in successive indentations (Fig. 1b–d and Supplementary Fig. 1). Similar results were found in tumors formed by the human breast adenocarcinoma MDA-MB-231 cells in mice (Fig. 1d). These findings establish the relevance of mechanical plasticity to breast cancer.

**IPNs with tunable mechanical plasticity**. Next, we designed a series of hydrogels for three-dimensional (3D) cell culture that exhibit differential plasticity while presenting the same initial stiffness and ligand density. These hydrogels consist of inter-penetrating networks (IPNs) of reconstituted BM (rBM) and alginate. The rBM network presents cells with ligands such as laminin and type IV collagen that are typically found in BM, while the tunable alginate network provides control over mechanical properties[15,16]. Alginate, an inert biopolymer derived from seaweed, presents no cell adhesion ligands, is not degradable by mammalian enzymes, and can be cross-linked ionically with divalent cations into a hydrogel[17]. The rBM and alginate were mixed and cross-linked to form IPN hydrogels with a final concentration of $4\,mg\,mL^{-1}$ rBM and $10\,mg\,mL^{-1}$ alginate (Supplementary Fig. 2a, b). Alginate molecular weight (MW) and cross-linker concentration were varied in order to form a set of IPNs with initial elastic moduli of around 1.8 kPa, similar to malignant breast tumors[12], but varying viscosity, as indicated by the loss tangent (Fig. 1e–g, Supplementary Fig. 2c, and Supplementary Table 1). We then quantified the plasticity of these IPNs, using creep and recovery tests to measure the degree to which each IPN's response to an applied stress was permanent (Fig. 1h). These IPNs demonstrated degrees of plasticity between 10 and 30%, for a 100 Pa creep stress applied for one hour, and from here on we refer to them as LP, medium plasticity (MP), and HP IPNs (Fig. 1i, j and Supplementary Fig. 2d). For comparison, covalently cross-linked polyacrylamide hydrogels exhibited a degree of plasticity of ~0%, whereas silly putty, a viscoelastic fluid, exhibited a plasticity of ~100%, indicating that even the HP IPNs are more similar to elastic hydrogels than they are to the malle-able silly putty (Fig. 1j). Unlike polyacrylamide, pure rBM and collagen-1 hydrogels, which are biologically relevant extracellular matrices (ECMs), exhibited significantly higher degrees of mechanical plasticity than the HP IPNs (Fig. 1k). The elastic–plastic yield stress of the HP IPNs was found to be below 10 Pa, and all IPNs behaved like viscoelastic solids, and not fluids, on time scales relevant to cellular migration events (Supplementary Fig. 2e, f). Matrix viscosity and plasticity are interrelated in this biomaterials system, as in many viscoplastic materials[18] (Fig. 1l). Gold nanoparticle diffusion experiments demonstrated that all IPNs had a pore size <40 nm, as 40 nm nanoparticles encapsulated into the different IPN formulations did not diffuse out of the gels (Fig. 1m). This pore size is on the order of BM pore size[1], and is well below the micron-sized pores thought to be required for cells to undergo known modes of protease-independent migration[5,7]. In summary, we have designed IPN hydrogels to offer varying degrees of plasticity, but equal concentrations of BM ligands, stiffness similar to that of tumor tissue, nanoporosity similar to BM, and limited susceptibility to cell-mediated degradation.

**Cells migrate in HP IPNs**. Using this materials system for 3D cell culture, we first investigated whether matrix plasticity affected breast cancer cell morphology in confining microenvironments. Highly invasive MDA-MB-231 cells were fully encapsulated in the IPNs and stimulated with $50\,ng\,mL^{-1}$ epidermal growth factor (EGF) (Fig. 2a). Striking morphological differences were observed between MDA-MB-231 cells in different IPNs after one day, with cells adopting circular morphologies in LP IPNs, highly protrusive morphologies in HP IPNs, and intermediate morphologies in the MP IPNs (Fig. 2b, c and Supplementary Fig. 3a). Differences in morphology due to plasticity were also found in HT-1080 fibrosarcoma cells, but not 4T1 or MCF7 cells (Fig. 2c). As the alginate network component of the IPN was nanoporous (Supplementary Fig. 2a, b), not degradable by pro-teases, and comprised ~70% by mass of the solid fraction of the IPN, changes in morphology were expected to be protease-independent. To confirm this, assays were repeated with the addition of GM6001, a broad-spectrum protease inhibitor, and results were similar to the vehicle-alone condition (Fig. 2d). To compare results from our 3D invasive morphology studies to those of an assay traditionally used for invasion, an adapted Boyden chamber invasion assay was performed (Supplementary Fig. 3b). MDA-MB-231 cells seeded on top of LP IPNs again exhibited rounded morphologies, whereas cells seeded on HP IPNs extended elongated, actin-rich protrusions into the IPN matrix below (Supplementary Fig. 3c). The results of these assays demonstrate that ECM plasticity regulates morphology of highly invasive cancer cells.

We next determined the impact of matrix plasticity on motility of the cancer cells. Time-lapse confocal microscopy and image analysis software were used to track the migration of RFP-LifeAct-transfected MDA-MB-231 cells encapsulated in the IPNs and stimulated with $50\,ng\,mL^{-1}$ EGF (Supplementary Movies 1–6). Strikingly, cells in HP IPNs could be regularly observed migrating through the nanoporous matrix (Fig. 2e and Supplementary Movie 1). In aggregate, cell track patterns suggest that cells in HP IPNs are more migratory than those in LP IPNs (Fig. 2f and Supplementary Movies 2, 3). Analysis of cell tracking data was used to estimate the likelihood that a cell in each of these IPNs would migrate, and revealed that cell migration probability was about five times higher among cells in HP IPNs than in LP IPNs (Fig. 2g). While the proportion of migrating cells differed significantly, the characteristics of migrating cells were similar (Supplementary Fig. 4). Control migration studies demonstrate that calcium, used to crosslink the IPNs, did not drive the observed changes in motility (Supplementary Fig. 5). As expected for cell migration in the predominantly alginate-based IPNs, protease inhibition with GM6001 or marimastat, another broad-spectrum protease inhibitor, did not diminish levels of cell motility (Fig. 2g, h, Supplementary Fig. 4, Supplementary Fig. 6a–e, and Supplementary Movies 4–9). Protease inhibition also failed to diminish cell motility in pure rBM, and in collagen-1 hydrogels without covalent crosslinks (Supplementary Fig. 6c, d). However, cell migration was limited in RGD (arginine–glycine–aspartic acid)-alginate hydrogels, suggesting that rBM ligands are important for activating invasive and migratory activities in the cells (Supplementary Fig. 6f)[19]. These assays reveal that cancer cells can migrate through nanoporous, confining matrices in a protease-independent manner when the matrices are sufficiently plastic.

**Cells extend invadopodia in HP IPNs**. After discovering a strong effect of matrix plasticity on cell migration, we sought to elucidate how cells initiated migration in HP matrices. For many of the motile cells in the HP IPNs, motility was preceded by the

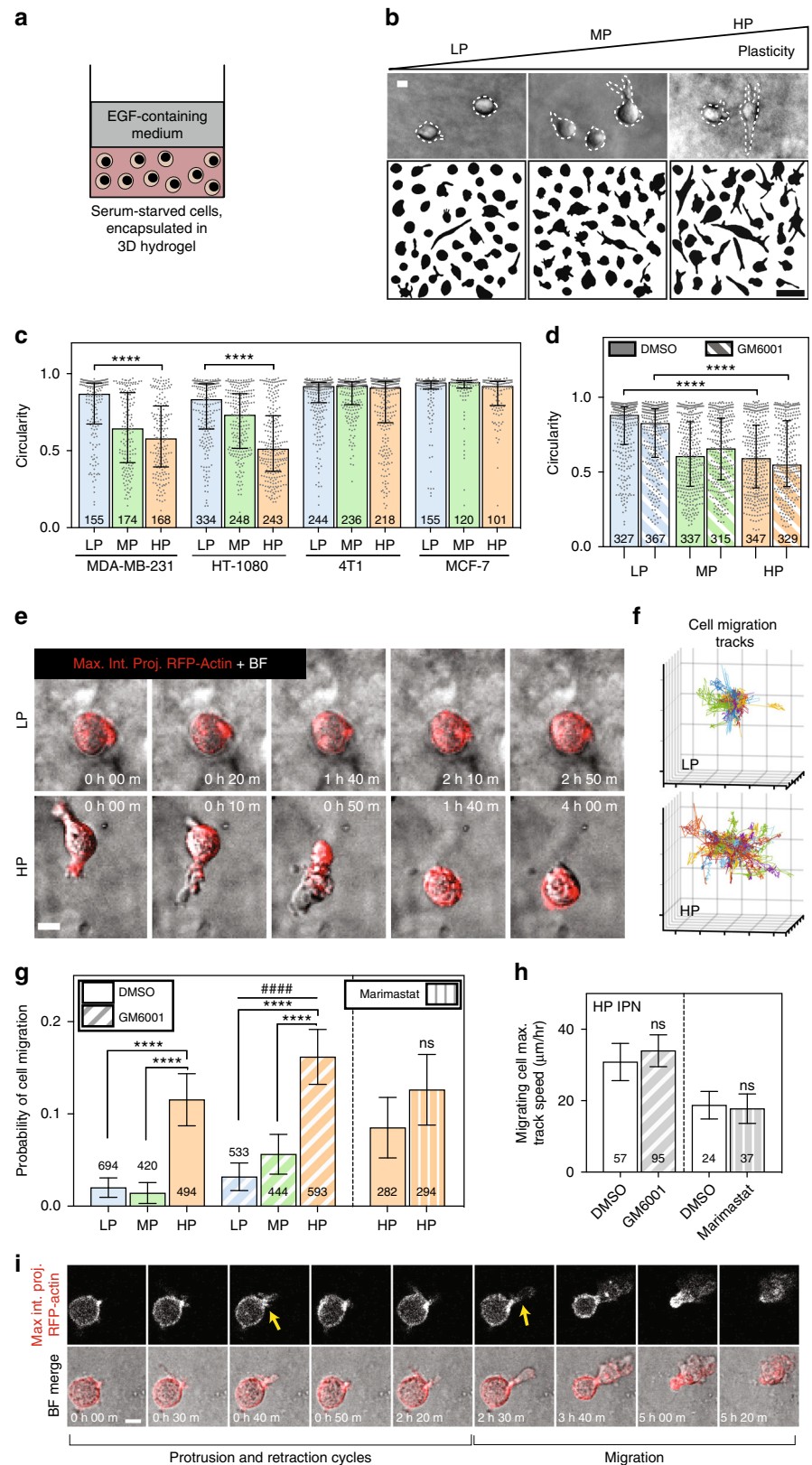

formation of oscillatory protrusions that resembled invadopodia (Fig. 2i, Supplementary Fig. 7a, and Supplementary Movies 1, 4). Invadopodia have lifetimes on the order of hours, evolve from puncta rich in β1 integrin and actin, display enriched cortactin (CTTN) and TKS5, and involve activation of β1 integrin, Arp 2/3 complex, and Rho GTPases[3,20,21]. Actin-rich puncta were present in cells in both LP and HP IPNs, but these puncta more frequently matured into elongated protrusions in HP IPNs (Fig. 3a and Supplementary Movies 10, 11). Cells in HP IPNs were ~50% more likely than cells in LP IPNs to extend protrusions (Fig. 3b and Supplementary Fig. 7b). Furthermore, protrusions in HP IPNs were longer (10–20 μm or more in length) and narrower

**Fig. 2** Enhanced matrix plasticity promotes spreading and motility of cancer cells independent of proteases. **a** Schematic of experimental setup. **b** After 1 day in 3D culture, cells were imaged using bright field microscopy and cell outlines were traced. Example MDA-MB-231 cells and cell outlines shown. Top scale bar is 10 μm, and bottom scale bar is 50 μm. **c** Circularity was calculated for traces of cells from four different cell lines. Data shown are from one representative biological replicate experiment. **d** Cell circularity was also quantified for MDA-MB-231 cells in the presence of broad-spectrum protease inhibitor (10 μM GM6001) or vehicle-alone control. Data shown are from two pooled experiments each. The validity of comparing medians of pooled data sets was verified (Supplementary Tables 2 and 3). For **c** and **d**, bars indicate median circularity of number of cells indicated, error bars indicate interquartile range, and statistical tests compared medians (****$P < 0.0001$, Mann–Whitney). **e** Time-lapse microscopy was used to image RFP-LifeAct-transfected MDA-MB-231 cells stimulated with 50 ng mL$^{-1}$ EGF. Maximum intensity projections of RFP-actin signal are shown merged with bright field images. Scale bar is 10 μm. **f** Representative 3D cell track reconstructions for cells in LP ($n = 143$) and HP ($n = 114$) IPNs. Grid size is 10 μm. **g** Probability of cell motility shown for LP, MP, and HP IPNs, with vehicle alone (DMSO) or protease inhibitor (10 μM GM6001 or 100 μM marimastat) added to the medium. Differences in mean probability of motility indicated are statistically significant (****$P < 0.0001$, Fisher's exact). Probability of motility trends with plasticity for protease inhibitor studies (####$P < 0.0001$, $\chi^2$ test for trend). **h** Maximum speeds for motile cells in HP IPNs, with vehicle-alone and protease inhibitor conditions (t tests; ns not significant). For both **g** and **h**, graph displays the number of cells analyzed per condition, taken from $R = 3$–5 biological replicate experiments, bars indicate mean probabilities, and error bars indicate 95% confidence intervals. **i** A representative cell in HP IPN exhibiting protrusion and retraction cycles prior to migrating. Scale bar is 10 μm

compared to those in LP IPNs (Fig. 3c, d and Supplementary Fig. 7a). The high aspect-ratio protrusions observed in HP IPNs also had longer lifetimes (hours), compared to the shorter, thicker protrusions commonly observed in LP IPNs (Fig. 3e and Supplementary Fig. 7c, d). In HP IPNs, actin-rich puncta reinforced many β1 integrin-rich plaques that excluded paxillin to their peripheries (Fig. 3f and Supplementary Fig. 7e). By perturbing key force-generating pathways, the protrusive phenotype was found to be dependent on β1 integrin, Rac1, and Arp 2/3 complex but not RhoA (Supplementary Fig. 7f). In addition, the roles of two canonical invadopodia markers, CTTN and TKS5, were investigated. In LP IPNs, CTTN–actin colocalization was limited (Fig. 3g) and TKS5 presence was diffuse (Fig. 3h). By contrast, in HP IPNs, CTTN localized both to protrusion precursors and to elongated protrusions (Fig. 3g), and spots of enriched TKS5 were observed at the bases of protrusions (Fig. 3h). Knockdown of either CTTN or TKS5 significantly decreased the frequency of the invasive protrusions (Fig. 3i and Supplementary Fig. 7g, h). Together, these findings establish the protrusions that form in HP IPNs to be invadopodia.

**Cells mechanically open up channels to migrate through.** Finally, we investigated the mechanisms underlying protease-independent cell migration through plastic matrices, testing the hypothesis that cells migrate through HP matrices by applying forces to generate openings in the matrix to migrate through. The probability of cell migration was significantly diminished by inhibition of Rac1, RhoA, Arp 2/3, myosin II, and F-actin (Fig. 4a, Supplementary Fig. 8, and Supplementary Movies 12–17), implicating molecular signaling pathways known to regulate both invadopodia and 3D cell migration, as well as a combination of cellular protrusivity and contractility, in the plasticity-dependent migration observed[3,20,22]. To determine the nature of force generation, fluorescent beads were embedded into the IPNs for cell migration studies in order to map matrix deformations resulting from forces generated by cell protrusions and motility (Supplementary Movie 18, Part 1). Matrix displacement maps reveal that in HP IPNs, invadopodia applied a combination of protrusive and contractile forces to deform the surrounding ECM as the invadopodia widened openings in the IPN (Fig. 4b and Supplementary Fig. 9a). Following these cycles of invadopodia extension and retraction, motile cells generated protrusive forces at the leading edge as they migrated (Fig. 4c, d and Supplementary Fig. 9b–e). Matrix displacement around migrating cells was significantly lower with inhibition of F-actin and myosin II (Fig. 4e). Maximum matrix displacements were larger in HP IPNs than MP and LP IPNs in magnitude, although differences were not significant

(Fig. 4f). As cells transitioned from invadopodia extension to migration, the wide and actin-rich leading edge often observed in migrating cells resembled lamellipodia, whose protrusive force-generating capabilities are known[8,23,24], though not previously observed in 3D (Fig. 4c and Supplementary Fig. 9b–d). Computational analysis of bead displacement maps indicate, on average, an upper bound of ~100 Pa for mechanical stress generated in the IPNs during migration (Supplementary Fig. 9d, e). To directly assess the nature and extent of plastic remodeling of the IPNs by migrating cells, cells were encapsulated in HP IPNs made with fluorescent alginate. Strikingly, matrix densifications were observed where protrusions extended or formerly extended, as well as along the edges of lasting channels through which cells migrated, which were on the order of 2–3 μm wide at their base and more than 50 μm long in length in some cases (Fig. 4g–i Supplementary Fig. 9f, g). These data show that cells mechanically remodel the IPNs, using protrusions to open up permanent channels through which they can subsequently migrate (Fig. 4h, i, Supplementary Fig. 9f, g, and Supplementary Movie 18).

## Discussion

Taken together, our data reveal that cell-generated forces, initiated by invadopodia, can facilitate protease-independent invasion and migration through confining microenvironments if the surrounding ECM is sufficiently plastic (Fig. 5). While cells can adopt various forms of protease-independent migration when confined to micron-sized rigid channels[7,25–27], pre-existing holes[10], and microtracks[11], migration through a 3D confining environment was previously thought to require protease degradation. The protease-independent mode of migration through confining matrices discovered here is initiated by invadopodia protrusions, which apply both protrusive and contractile forces to initiate opening of the matrix. Previously, ECM degradation was considered to be the primary function of invadopodia, with prior studies finding that invadopodia generate forces in order to aid degradation[4,28]. However, our studies reveal that invadopodia can generate force and physically expand pores, independent of their role in degradation. Our approximations of stresses generated by protrusive and migrating cells, as well as the protrusive and traction stresses measured by others (1–10 kPa)[24,29], are well beyond the yield stress of the HP IPNs (below 10 Pa), so it is expected that cells would generate forces that are sufficient to permanently deform these HP IPNs. However, the ability of invadopodia to act through degradation and force may be cell type-dependent. While our study and others indicate that protease-inhibited and untreated MDA-MB-231 cells show similar invadopodia characteristics[21], 4T1 cells were unable to

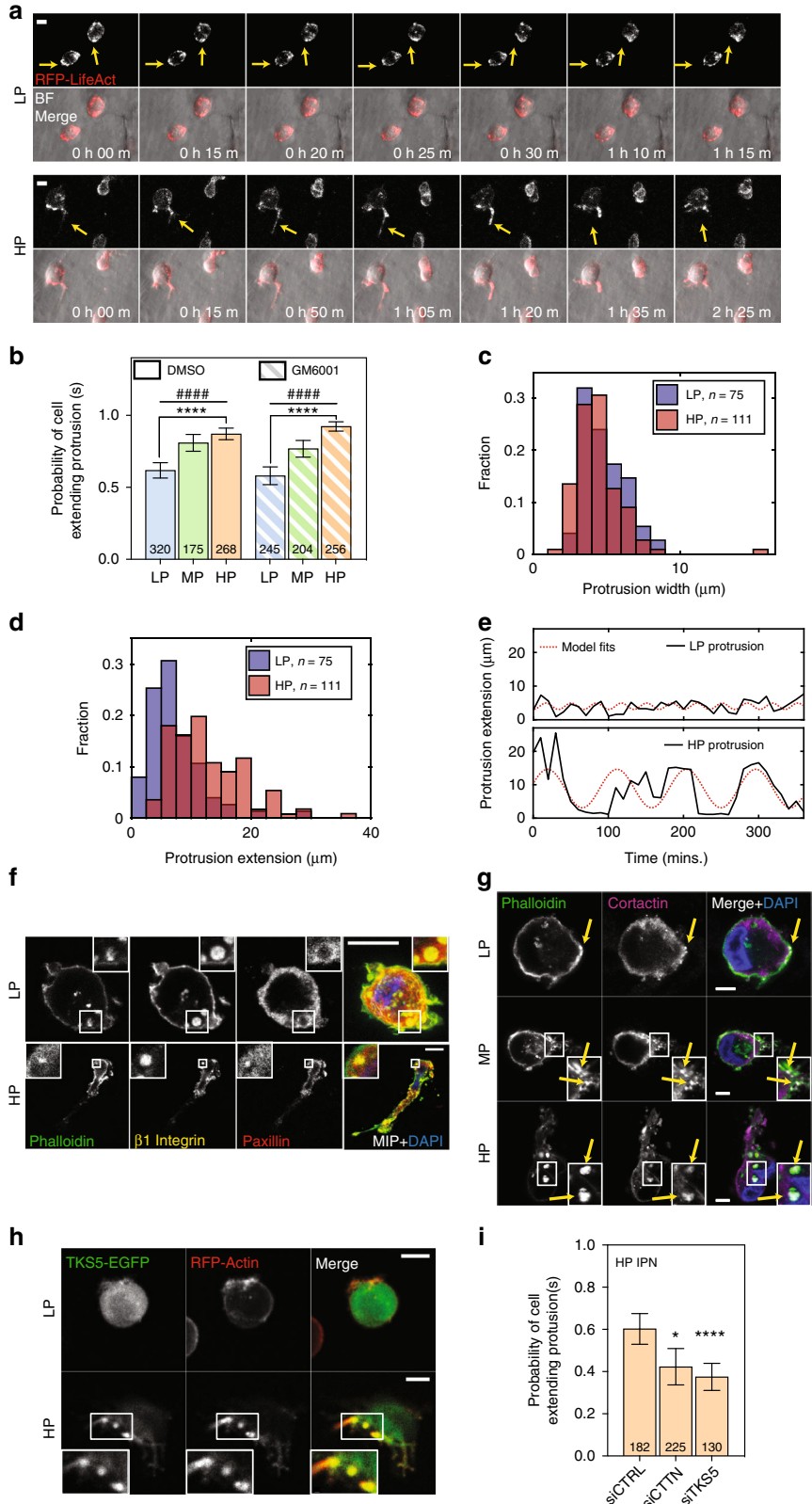

form protrusions in the HP IPNs, even though they are known to form invadopodia that robustly degrade matrix[30]. The oscillatory nature of the protrusions in HP IPNs, which typically lengthened and widened leading up to cell migration, suggests that these structures widen pores in stages. This observation underlies the importance of the matrix being plastic enough to accumulate and

sustain deformations over time. To this point, the finding that cell morphologies in HP and MP IPNs are similarly protrusive but the likelihood of cell migration in these matrices is significantly different suggests a threshold matrix plasticity required for channels formed by invadopodia to stay open wide enough to enable cell migration. Cells migrated after invadopodia protrusions formed

**Fig. 3** MDA-MB-231 cells in high plasticity matrices extend invadopodia protrusions. **a** RFP-LifeAct MDA-MB-231 cells were imaged for 12 h using time-lapse confocal fluorescence microscopy. Transient actin-rich spots were observed in all IPNs (yellow arrows, top), but in HP IPNs, these spots often elongated into oscillatory, actin-rich protrusions (yellow arrows, bottom). Scale bar is 10 μm. **b** Probability of a cell extending one or more protrusions in LP, MP, and HP IPNs, with either vehicle alone or protease inhibitor (10 μM GM6001) added to the medium. Graph displays the number of cells per condition, taken from $R = 3$–5 biological replicate experiments. Bars indicate mean probabilities and error bars indicate 95% confidence intervals. Differences in protrusivity as indicated are statistically significant (****$P < 0.0001$, Fisher's exact). Probability of extending a protrusion also trends with plasticity (####$P$ $< 0.0001$, $\chi^2$ tests for trend). **c** and **d** Histograms of extensions and widths of cell protrusions in LP and HP IPNs. Includes number of cells indicated per condition, pooled from $R = 3$ biological replicate experiments each. **e** Tracings of protrusion lengths, for one cell each in LP and HP IPNs, over time during a 6 h timeframe, plotted alongside a sinusoidal fit. **f**–**h** Confocal immunofluorescence imaging was used to investigate localization of indicated proteins. Main panel scale bar is 10 μm. **f** and **g** Imaging of indicated staining on cryosections of MDA-MB-231 cells encapsulated in IPNs for 1 day. In **f**, main image shows merged maximum intensity projection (MIP) and inset (1.5× zoom for LP and 4× zoom for HP) shows merged image on one z-plane. In **g**, inset is 2× zoom. **h** Live imaging of actin (RFP-LifeAct) and TKS5 (TKS5-EGFP). Inset is 1.5× zoom. **i** Probability of a cell extending one or more protrusions, for cells transfected with control siRNA, siRNA targeting cortactin (siCTTN), or siRNA targeting TKS5 (siTKS5)

channels on the order of several microns wide, consistent with the previously established idea that a stiff nucleus can only squeeze to a size of several microns via Rho-mediated actomyosin contractility, and therefore the nucleus impedes cell migration until such a size opening becomes available[5,7]. Indeed, Rho-mediated actomyosin contractility was required for migration but not protrusivity. Though the number of cells that migrated was highly dependent on matrix plasticity, the few cells that did migrate in LP or MP IPNs migrated distances similar to those in HP IPNs, and at speeds consistent with 3D cancer cell invasion in other material systems, which report speeds ranging from 2–40 μm h$^{-1}$ [5,20,31,32]. These data are consistent with the explanation that what dictates the speed and distance is the length of the channel formed by the invadopodial protrusion. Under this explanation, the cells do not migrate until a channel of a sufficient width and distance is formed, and then, once such a channel is formed, migrate the length of that channel. The observation that cell morphology and migration likelihood depend on β1 integrin and Arp 2/3, but migration speed and distance do not, suggests that channel formation and migration through the channel may be decoupled mechanistically. The observation that invadopodia are utilized in both protease-dependent migration and this plasticity-mediated mode of protease-independent migration could indicate that these modes represent two extremes of mesenchymal migration. Altogether, these data demonstrate a previously unreported mode of protease-independent cell migration through nanoporous matrices that is mediated by matrix plasticity.

There is strong indication for the in vivo relevance of this new mode of migration. While the mode of migration described here has not directly been identified in vivo, previous work finding that invadopodia can physically displace BM during development in *Caenorhabditis elegans*[33] could be explained by the BM exhibiting plasticity. Direct evidence for this mode of migration in mouse models is currently missing, as it is challenging to unambiguously assess the independent contributions of proteases vs. mechanical plasticity in mouse models of BM invasion by primary tumors. However, the elevated viscosity of breast tumor tissue that has been observed clinically[13], the aberrant cross-linking and matrix architectures associated with tumor progression[14], and our data revealing substantial mechanical plasticity of human and mouse tumor tissue all point toward the likelihood of both the BM and stromal matrix exhibiting some degree of mechanical plasticity during breast cancer. As the plasticity levels required for migration are relatively low, with the HP IPNs exhibiting a degree of plasticity of only 30% in response to a creep stress of 100 Pa applied for 1 h, these findings support the strong likelihood that plasticity-mediated migration is relevant to breast cancer invasion and migration.

Though our study focused on breast cancer cells, this new mode of invasion and migration could be broadly relevant to and synergistically act with other modes of migration. This mode could be harnessed by other cells that cross BM during physiological processes, such as immune cells[34], or aided by other cells that infiltrate the tumor microenvironment, such as cancer-associated fibroblasts[14,33]. Indeed, a recent study found that cancer-associated fibroblasts can dilate pre-existing gaps in the BM to facilitate carcinoma cells crossing the BM[10]. Further, this mode of migration may be utilized by cells to migrate through other confining microenvironments, such as the col-1-rich stromal matrix in breast tissue, where collagen density often increases during breast cancer progression[11,35]. In vivo, cells may utilize a combination of degradation and force generation on BM and stromal matrix to synergistically enable robust invasion of confining microenvironments, and then transition to a faster ameboid mode of protease-independent migration as they migrate into microenvironments with larger pore sizes, enabled by aligned collagen networks and fiber bundling[36,37].

The evidence that HP ECM enables breast cancer cell invasion independent of proteases in vitro may provide new insight into poor outcomes of protease inhibition clinical trials. Clinical trials have generally concluded that broad-spectrum protease inhibitors employed therapeutically cause substantial off-target effects and provide little to no survival benefit to cancer patients[38,39]. Our data support the investigation of pharmacological interventions that simultaneously perturb both protease-dependent and protease-independent invasion processes. For example, it is possible that the combination of a protease inhibitor with an inhibitor of a force-generating pathway or protein, such as RhoA, Rac1, or Arp 2/3, may prove to be more effective than protease inhibition alone in blocking invasion. Overall, while mechanobiological studies of invasion and migration to date have focused on the impact of ECM stiffness and matrix architecture[8,40], these results establish plasticity, a mechanical property distinct from stiffness, as an important regulator of cancer cell invasion and metastasis, adding to our evolving understanding of how cells negotiate confining 3D environments.

## Methods

**Mechanical testing of human and MDA-MB-231 tumors**. Human breast carcinoma specimens were obtained from the Stanford Tissue Bank. An International Review Board waiver was obtained for these studies, as the specimens were excess tissue not collected for the current research, and were de-identified prior to use. Specimens were stored in serum-free RPMI at 4 ℃ prior to testing. All animal experiments were done according to a protocol approved by the institutional animal care and use committee. Mammary tumors were induced in 8–10-week-old NU/NU female mice (Charles River Laboratories). The mice were inoculated with $1 \times 10^6$ MDA-MB-231 cells, suspended in 50 μL of phosphate-buffered saline (PBS), in the right inguinal mammary gland in proximity to the fourth nipple.

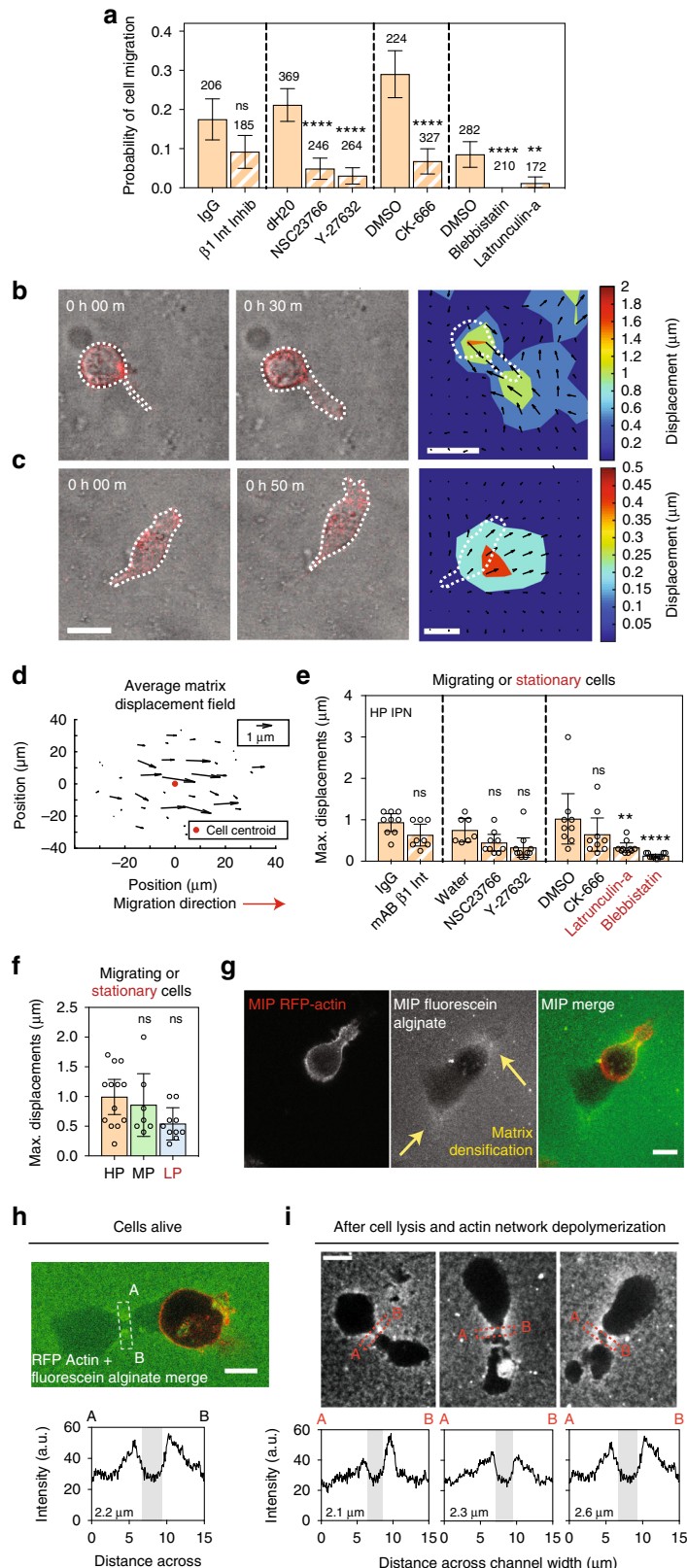

After 21 days, tumors from four separate mice were resected and sectioned into 2-mm-thick discs with a 6 mm diameter for mechanical testing.

Tissue mechanical plasticity testing was performed using an Instron 5848 material testing system with a 1 N load cell (Futek). Tissue samples were cut into 6 mm diameter plugs, submerged in serum-free RPMI medium, and then were equilibrated to room temperature prior to testing. For mechanical testing, unconfined compression indentations of ~40% of the specimen thickness, at a

normalized strain rate of 0.5 mm min$^{-1}$, were applied while the plug was submerged in serum-free RPMI. For human tumor tissue testing, a 4 mm indenter was used to apply repeated indentations that were held for 1 h and allowed to recover for 1 h. For MDA-MB-231 tumor tissue testing, a 1.5 mm indenter was used to apply repeated indentations that were held for 1 h and allowed to recover for 2 h. Because the initial point of contact could be difficult to determine from force vs. indentation curves for the tissue plug, indentation depth at 25% of the

**Fig. 4** In highly plastic ECM, cell-generated forces displace the matrix plastically to facilitate invasion and migration. **a** Probability of cell migration, with the indicated vehicle alone or inhibitor, added to the media. Drug/antibody concentrations used to inhibit/block respective pathways were: 1 µg mL$^{-1}$ monoclonal β1 integrin-blocking antibody (β1 integrin), 70 µM NSC23766 (Rac1), 10 µM Y-27632 (ROCK), 100 µM CK-666 (Arp 2/3), 50 µM Blebbistatin (myosin II), and 2.5 µM Latrunculin-a (F-actin). Cells from $R = 3$ biological replicate experiments, bars indicate mean probabilities, and error bars indicate 95% confidence intervals. **b, c** Images from confocal time-lapse studies of representative RFP-LifeAct MDA-MB-231 cells, encapsulated with fluorescent beads and stimulated with 50 ng mL$^{-1}$ EGF, depicting **b**, a cell extending a protrusion, and **c**, a cell migrating. Bead displacements obtained from a single $z$-plane and time points shown were used to inform models of the matrix displacement field, which is superimposed over a heat map illustrating displacement magnitudes and directions as calculated. Scale bar is 20 µm. **d** Average matrix displacement map, which incorporates information from $N = 16$ cells from $R = 3$ biological replicate experiments. All centroids were located at the red indicator, and all maps were rotated such that the cells migrated to the right. **e, f** Maximum bead displacements observed around migrating cells for conditions in which cell migration was observed, and around stationary cells when cell migration was absent. **e** Maximum bead displacements around cells in HP IPNs for indicated conditions. Statistically significant differences are indicated (**$P < 0.01$, ****$P < 0.0001$, ANOVA; ns not significant). **f** Maximum bead displacements around cells in HP, MP, and LP IPNs (ANOVA). **g, h** Cells in HP IPNs made with fluorescein-conjugated alginate. **g** Matrix is densified around protrusions (top yellow arrow), and densification persists after protrusions retract (bottom yellow arrow). **h** Migrating cells leave lasting channels. The fluorescence intensity signal across the migrating cell's path (line marked A to B) was measured. **i** Three hours after cells were lysed and actin networks were depolymerized, similar channels remained, with their intensity profiles as shown. For **g–i**, scale bar is 10 µm

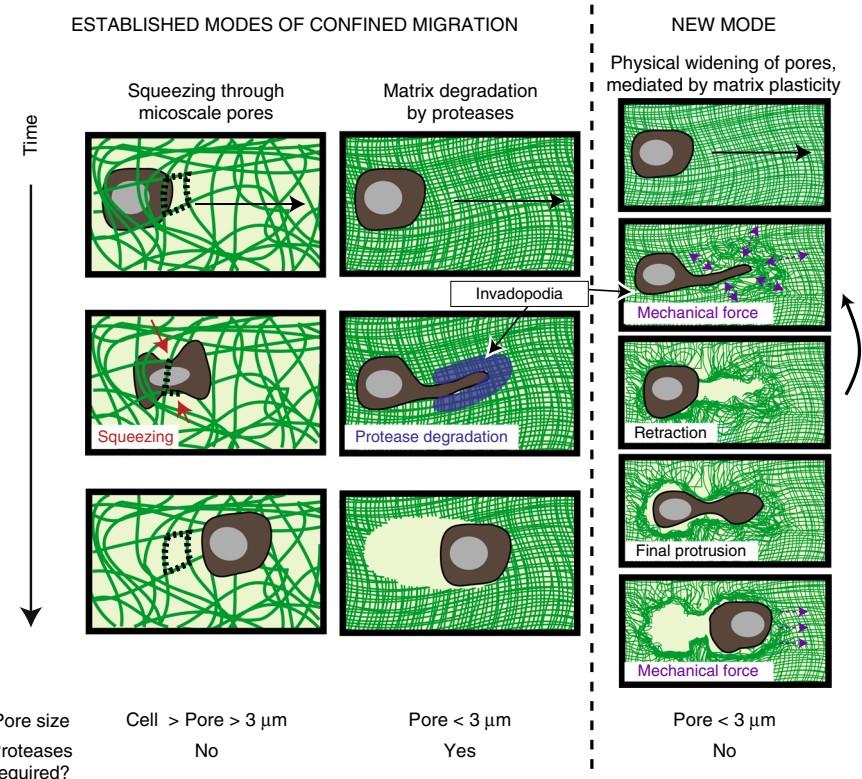

**Fig. 5** Known modes and newly discovered mode of confined migration. For pores sizes smaller than a cell but larger than ~3 µm, it is thought that cells can squeeze through pores to migrate, without requiring proteases. It is thought that for pores smaller than ~3 µm, cells are considered confined to such a degree that they require proteases to migrate. We report a migration mode that is plasticity-mediated and protease-independent: if pores are smaller than ~3 µm and the matrix is sufficiently plastic, then cells can use progressively widening and lengthening protrusions to physically open up a channel in the surrounding matrix and enable cell migration

initial peak force was used to evaluate the indentation plasticity between indentation cycles. Peak forces were also compared between successive indentation cycles.

**Alginate preparation**. Sodium alginate rich in guluronic acid blocks and with a high-MW (FMC Biopolymer, Protanal LF 20/40, High-MW, 280 kDa) was prepared[41]. High-MW was irradiated 3 or 8 Mrad (3 or $8 \times 10^6$ rad) by a cobalt source to produce mid-MW (70 kDa) and low-MW (35 kDa) alginates[16]. Fluorescein-coupled alginate was prepared by coupling fluoresceinamine isomer (Acros Organics) to the alginates using carbodiimide chemistry at a concentration of 37.74 µM. Alginate was dialyzed against deionized water for 3–4 days (MW cutoff of 3500 Da), treated with activated charcoal, sterile-filtered, lyophilized, and then

reconstituted in serum-free Dulbecco's modified Eagle's medium (DMEM) (Life Technologies).

For preparation of Low-MW RGD-alginate, oligopeptide GGGGRGDSP (Peptide 2.0) was coupled to the Low-MW (35 kDa) alginate using standard carbodiimide chemistry[42]. In a typical reaction, 1 g of alginate was reconstituted at 1% wt vol$^{-1}$ in MES (2-(N-morpholino)ethanesulfonic acid) buffer (0.1 M MES, 0.3 M NaCl, pH 6.5). For 1% wt vol$^{-1}$ alginate IPNs containing 750 µM RGD, the following procedure was used: 35.9 mg of sulfo-NHS (Thermo Fisher Scientific), 63.47 mg of N-(3-dimethylaminopropyl)-N'-ethylcarbodiimide (Sigma) and 56.8 mg of peptide were added, and then the reaction proceeded for 20 h before quenching with hydroxylamine hydrochloride (Sigma). Alginate was then dialyzed, charcoal-treated, sterile-filtered, lyophilized, and reconstituted as described above.

**Mechanical characterization of IPNs.** Rheology measurements were made with an AR2000EX stress-controlled rheometer (TA Instruments). IPNs, which were made for mechanical testing, were deposited directly onto the bottom plate of the rheometer immediately after mixing with cross-linker. A 25 mm flat plate was then immediately brought down, forming a 25 mm disk of gel. Mineral oil (Sigma) was applied to the edges of the gel disk to prevent dry-out. The mechanical properties were then measured over time until the storage modulus reached an equilibrium value. The storage and loss moduli at 1 rad s$^{-1}$ and 1% strain, a frequency and amplitude which were both within linear regimes, were recorded periodically for at least 2 h. Elastic moduli (i.e., Young's moduli) were calculated assuming a Poisson's ratio of 0.5 using the equation:

$$E = 2(1 + v)G^*, \qquad (1)$$

where $G^*$ is the complex modulus found using the storage and loss moduli measured, calculated using

$$G^* = (G'^2 + G''^2)^{1/2}. \qquad (2)$$

For plasticity experiments, this time sweep was followed by a creep-recovery test. This involved first applying a constant shear stress (10, 50, 100, or 150 Pa) for 1 h, while strain was recorded as a function of time. Then, the sample was unloaded (0 Pa) and strain was recorded as a function of time as the sample recovered from the absence of load for 6400 s (1.7 h). This recovery time period was sufficient to minimize transient effects due to stress unloading, and was on the same time scale as the periodic structures and migration events observed in this study. To establish that the hydrogels behave as viscoelastic solids over time scales relevant to these cellular behaviors, additional stress relaxation tests at low strain (5% strain) were conducted.

**Interpenetration characterization of IPNs.** To confirm interpenetration of rBM and alginate networks, fluorescence microscopy of IPNs formed with fluoresceinamine-coupled alginate were performed. Gels were imaged 1 day after formation, using a Leica SP8 confocal microscope and a Leica HC PL APO ×63/1.4 NA oil immersion objective. ImageJ (US National Institutes of Health, Bethesda, MD, USA, https://imagej.nih.gov/ij/, 1997–2016) was used to measure the fluorescence intensity of all pixels in 10 images of each of the three separate IPNs. Intensity values were then displayed as histograms. The single-peaked and narrow fluorescence intensity histograms indicate that the alginate is present in each pixel, and thus that the alginate and rBM networks are interpenetrating, within the resolution of images taken (~200 nm). Resolution was quantified using the Rayleigh criterion, where resolution is estimated as $\lambda \left(2 \times NA_{obj}\right)^{-1}$.

**Pore size estimation of IPNs.** To determine the approximate pore size of the IPN hydrogels, gold nanospheres of various sizes pre-coated with 2 kDa PEG (Nano-Hybrids, Austin, TX, USA) were encapsulated into all IPN hydrogels[15]. Phenol red-free DMEM with 1% penicillin/streptomycin (Pen/Strep) was added on top of the hydrogels, and samples were incubated at 37 °C and 5% CO$_2$. Samples were taken out of the incubator and gently shaken for 10 min every 1–2 days. After 4 days, the final supernatant volume was quantified, and the supernatant absorbance was measured using a plate reader. The particle concentration was calculated using the absorbance at the provided batch-specific wavelength of maximum absorbance. The relative diffusion of the particles out of the IPNs was estimated using the partition coefficient. The partition coefficient was then quantified as the concentration of particles in the supernatant divided by the concentration of particles remaining in the hydrogel, which was approximated using both the concentration of nanoparticles originally encapsulated and the concentration of nanoparticles detected in the supernatant post-diffusion. A partition coefficient of 1 indicates equal concentration of nanoparticles in the hydrogel and the supernatant; a partition coefficient close to zero indicates that particles remain trapped in the hydrogel. The estimation of a pore size of <40 nm is consistent with a previous study, which estimated the pore size of a similar IPN formulation, though one with a lower concentration of alginate, to be ~30 nm by measuring the diffusion coefficient of molecules with a known MW[15].

**Cell culture.** Human breast adenocarcinoma cells MDA-MB-231 (ATCC), invasive ductal carcinoma cells MCF7 (ATCC), and fibrosarcoma cells HT-1080 (ATCC) were cultured in high glucose DMEM (Hyclone) with 10% fetal bovine serum (FBS) (Hyclone) and 1% Pen/Strep (Life Technologies). Mouse metastatic breast cancer cells 4T1 (ATCC) were cultured in RPMI (Hyclone) with 10% FBS and 1% Pen/Strep. All cells were cultured at 37 °C in 5% CO$_2$. MDA-MB-231 cells were stably transfected to express RFP-LifeAct using a Piggybac vector construct with Geneticin resistance (gift from A. Dunn). Cells were transfected with expression constructs using Lipofectamine (Invitrogen). Cells were analyzed after 24 h of transfection. The MDA-MB-231 cells stably expressing RFP-LifeAct were selected with 800 µg mL$^{-1}$ G418 and clones were expanded. RFP-LifeAct MDA-MB-231 cells were transiently transfected to express TKS5-EGFP using Polyjet[43] (SignaGen Laboratories). MDA-MB-231, MCF7, and HT-1080 cells were authenticated by the ATCC and tested to be mycoplasma negative.

For CTTN and TKS5 (SH3PXD2A) knockdowns, MDA-MB-231 cells stably expressing LifeAct-RFP were transfected with 50 nM CTTN small interfering RNA (siRNA) SMARTpool (cat. #M-010508-00-0005; Dharmacon), SH3PXD2A siRNA SMARTpool (cat. #M-006657-02-0005; Dharmacon), or ON-TARGETplus Non-Targeting Control Pool (cat. #D-001810-10-05; Dharmacon) using DharmaFECT 1 Transfection Reagent (cat. #T-2001-01; Dharmacon). Seventy-two hours following transfection, cells were assayed for knockdown by Western blot and imaged for protrusions. For Western blotting, MDA-MB-231::LifeAct-RFP/siRNA cells were harvested and centrifuged at $500 \times g$ for 5 min. Cell pellets were washed with serum-containing growth medium to neutralize trypsin and washed with PBS. For sodium dodecyl sulfate-polyacrylamide gel electrophoresis of whole cell lysates, cells were lysed in Pierce RIPA buffer (cat. #89900; Thermo Fisher Scientific) supplemented with Protease Inhibitor Cocktail Tablets (cat. #11836170001; Roche) and PhosSTOP Phosphatase Inhibitor Cocktail Tablets (cat. #04906845001; Roche) according to the manufacturer's instructions. Protein concentration was determined using the Pierce BCA Protein Assay Kit (cat. #23227; Thermo Fisher Scientific). Laemmli sample buffer (cat. #1610747; Bio-Rad) was added to lysates and samples boiled for 10 min before loading 25 µg protein in each lane of a 4–15%, 15-well, gradient gel (cat. # 4561086; Bio-Rad). Proteins were transferred to nitrocellulose at 100 V for 45 min, blocked with 5% milk in TBS-T (137 mM NaCl, 2.7 mM KCl, 19 mM Tris base, 0.1% Tween, pH 7.4), incubated overnight in primary antibodies against CTTN (1:1000; cat. #ab33333; Abcam) and SH3PXD2A (1:200; cat. #sc-376211; Santa Cruz Biotechnology), and p38 (1:2000; cat. #sc-535; Santa Cruz Biotechnology) was used as a loading control. Blots were incubated in IRDye 680-conjugated or 800-conjugated secondary antibodies (Li-COR Biotechnology) for 1 h and visualized using the Li-COR Odyssey imaging system (Li-COR Biotechnology). Quantitative analysis of western blots was performed using the Li-COR Odyssey software (LI-COR Biotechnology).

**3D cell encapsulation in IPNs.** For analysis of invasive morphology, all cancer cells (MDA-MB-231, HT-1080, 4T1, and MCF7) were starved overnight in serum-free medium and encapsulated in IPNs. In brief, cells in flasks were starved overnight prior to encapsulation[15,44]. They were then washed with PBS, trypsinized using 0.05% trypsin/EDTA, washed once, centrifuged, and resuspended in serum-free medium. The concentration of cells was determined using a Vi-Cell Coulter counter (Beckman Coulter). After Matrigel was mixed with alginate, cells were added into this polymer mixture and deposited into a cooled syringe. The solution was then vigorously mixed with a solution containing CaSO$_4$ and deposited into wells of a chambered coverglass (LabTek). The final concentration of cells was $0.5 \times 10^6$ cells mL$^{-1}$ of IPN. The cell-laden hydrogels gelled in an incubator at 37 °C and 5% CO$_2$ for 35–45 min, and then were stimulated with medium containing 10% FBS and 50 ng mL$^{-1}$ EGF. After one day, bright field microscopy was used to capture cell morphologies.

For time-lapse studies using RFP-LifeAct MDA-MB-231 cells in IPNs, a procedure similar to above was used. For migration studies, cells were encapsulated in IPNs at a concentration of $2 \times 10^6$ cells mL$^{-1}$ of IPN. After IPN gelation, FluoroBrite starvation medium was added: FluoroBrite (Thermo Fisher Scientific) supplemented with GlutaMAX (Thermo Fisher Scientific) and 1% Pen/Strep, with vehicle alone or inhibitor. One day later, starvation medium was removed; 60 µL of a 0.75% agarose (Sigma) hydrogel was added to each well to hold the IPN in place; and FluoroBrite invasion medium was added (FluoroBrite supplemented with GlutaMAX, 1% Pen/Strep, 15% FBS, 50 ng mL$^{-1}$ EGF, and either vehicle or inhibitor). Cells were then imaged live in an incubated chamber (37 °C and 5% CO$_2$) at 10 min (protease inhibitor studies) or 20 min (all other inhibitor studies) intervals with a Leica HCX PL APO ×10/0.40 NA objective overnight. For studies imaging RFP-LifeAct MDA-MB-231 protrusion dynamics, a similar procedure was followed, except for the following changes: the final concentration of RFP-LifeAct cells was $1.5 \times 10^6$ cells mL$^{-1}$ of IPN; cells were fed with FluoroBrite invasion medium (containing 10% FBS and 50 ng mL$^{-1}$ EGF) immediately after encapsulation; and cells were live imaged at 5 min intervals with a Leica HC FLUOTAR L ×25/0.95 water immersion objective. For time-lapse studies used to determine matrix displacement fields, RFP-LifeAct MDA-MB-231 cells were encapsulated with 0.2 µm fluorescent microspheres (Thermo Fisher Scientific) at a density of $1.6 \times 10^{12}$ beads mL$^{-1}$ [31]. For matrix visualization studies, RFP-LifeAct MDA-MB-231 cells were encapsulated in IPNs formed using fluorescein-conjugated alginate, and imaged using a Leica HC PL APO ×63/1.4 NA oil immersion objective. For matrix visualization studies after cell lysis, hydrogels were additionally incubated in medium containing 1% Triton X-100 and 50 µM cytochalasin D for 3 h prior to imaging the hydrogels.

For studies using cells transfected with TKS5-EGFP or siRNA studies (siCTTN, siTKS5, or siControl), a similar time-lapse procedure was followed. Cells were expanded for 1 day (TKS5-EGFP) or 2 days (siRNA studies) in complete growth medium. Cells were then encapsulated in IPNs at a concentration $0.8 \times 10^6$ cells mL$^{-1}$ (TKS5-EGFP) or $2 \times 10^6$ cells mL$^{-1}$ (siRNA studies). Then, cells were starved overnight in FluoroBrite starvation medium. Prior to imaging, starvation medium was removed; 60 µL of a 0.75% agarose (Sigma); and then FluoroBrite invasion medium (FluoroBrite supplemented with GlutaMAX, 1% Pen/Strep, 15% FBS, and 50 ng mL$^{-1}$ EGF) was added to each well. Cells were live imaged in an incubated chamber (37 °C and 5% CO$_2$) with a Leica HCX PL APO ×10/0.40 CS objective.

We note that control conditions were included in each and every time-lapse experiment alongside the relevant experimental conditions. The mark-and-find

feature of the Leica software was used to image multiple samples at every time point. This approach provided internal controls for cell passage number and environmental conditions within each individual experiment.

**Modified Boyden chamber invasion assay.** An adapted Boyden chamber invasion assay was used. MDA-MB-231 cells were starved overnight in serum-free DMEM. IPN hydrogels were deposited on top of a Snapwell insert (Corning® Costar®) and gelled for 35–45 min at 37 °C in 5% $CO_2$. Cells were washed with PBS, trypsinized with 0.05% trypsin/EDTA (Life Technologies), washed with serum-free DMEM, centrifuged, and resuspended in serum-free medium. Cells ($10^5$) were plated on top of the IPN in serum-free medium. Chemoattractant medium containing 10% FBS and 50 ng mL$^{-1}$ EGF was placed at the bottom of the chamber. Chemoattractant medium was replaced, and the plate was gently shaken on an orbital shaker every 1–2 days for 9 days. Hydrogels were then prepared for immunohistochemistry as described below, and then imaged using confocal immunofluorescence.

**Gelatin degradation assay.** Alexa-405 NHS-ester (Invitrogen) was conjugated to 0.2% porcine gelatin (Sigma) in PBS following the manufacturer's protocols. Twenty-five millimeter, #1.5 circular glass coverslips (Warner) were coated with a thin layer of fluorescent gelatin[45]. Briefly, coverslips were first coated with poly-L-lysine (50 µg mL$^{-1}$) for 20 min, washed three times with PBS, and then incubated with 0.02% glutaraldehyde in PBS for 15 min, then washed three times with PBS, and finally incubated with pre-warmed (37 °C) fluorescently labeled gelatin for 10 min. Coverslips were washed three times in PBS followed by quenching of glutaraldehyde using 5 mg mL$^{-1}$ solution of sodium borohydride (Sigma) for 15 min. Coverslips were washed three times in PBS and kept in 35 mm cell culture dishes in PBS containing penicillin (100 IU mL$^{-1}$) and streptomycin (100 µg mL$^{-1}$) at 4 °C until use. Prior to plating cells, dishes containing coverslips were pre-incubated in normal culture medium for 30 min. Cells were plated at a density of $1 \times 10^5$ on these coverslips, either with 10 µM GM6001 in dimethyl sulfoxide (DMSO) or DMSO vehicle alone (final DMSO concentration was 0.04%), for 16 h at normal cell culture conditions. Cells were fixed in 3.7% paraformaldehyde, permeabilized with 0.3% Triton X-100, and stained for CTTN (mouse; Abcam ab33333) and TKS5 (rabbit; Santa Cruz M-300). Gelatin degradation was measured by quantifying the average area of non-fluorescent pixels per field using a manual threshold in ImageJ. Twenty random fields were imaged per condition per experiment, and each independent experiment was performed three times. Invadopodia were identified by co-staining with CTTN and TKS5 antibodies and manually counted from images.

**3D migration control studies.** To verify that alginate is not degraded in our studies, cells were encapsulated into 1% alginate hydrogels at a concentration of $2 \times 10^6$ cells mL$^{-1}$. Half of the samples were immediately placed into micro-centrifuge tubes, frozen, and lyophilized, and the other half of the samples were cultured for one day in FluoroBrite starvation medium, and then for another day in invasion medium containing 50 ng mL$^{-1}$ EGF, similar to the procedure for time-lapse 3D invasion assays in IPNs. The second half of the samples were then placed in microcentrifuge tubes, frozen, and lyophilized. After samples were fully freeze-dried (2–3 days), they were weighed.

To control for the effect of soluble calcium on the migratory ability of RFP-LifeAct MDA-MB-231, cells were seeded in 4.0 mg mL$^{-1}$ collagen at a concentration of $1.5 \times 10^6$ after an overnight serum-free FluoroBrite starvation medium. After collagen gelation, FluoroBrite was supplemented with GlutaMAX and 1% Pen/Strep with 0, 9, or 21 mM of calcium. While these are the same concentrations of calcium used to crosslink the LP and HP IPNs, we note that the soluble calcium is likely lower in the IPNs. Cells were then stimulated with invasion medium containing 50 ng mL$^{-1}$ EGF and imaged live in an incubated chamber (37 °C and 5% $CO_2$) at 20 min intervals with a Leica HCX PL APO ×10/0.40 NA objective overnight.

To determine the importance of the rBM component of the IPNs, RFP-LifeAct MDA-MB-231 cells were encapsulated in 1% wt vol$^{-1}$ RGD-alginate hydrogels at a concentration of $2 \times 10^6$ cells mL$^{-1}$. Cells were imaged live in an incubated chamber (37 °C and 5% $CO_2$) at 20 min intervals with a Leica HCX PL APO ×10/0.40 NA objective overnight.

To analyze the effect of matrix metalloproteinase (MMP) inhibitors on the migration of RFP-LifeAct MDA-MB-231 in collagen and rBM gels, cells were encapsulated in collagen or rBM gels with or without protease inhibitor. Collagen gels were prepared using High Concentration Rat Tail Collagen (Corning, Product #354249) according to the manufacturer's protocol. Because this commercially available collagen-1 product does not contain intact telopeptides, the collagen-1 hydrogel is not covalently cross-linked. The collagen solution was adjusted to the desired pH with NaOH and mixed with 10x PBS to the desired concentration of 4.0 mg mL$^{-1}$. Cells were encapsulated at a concentration of $1.5 \times 10^6$ cells mL$^{-1}$ in 4.0 mg mL$^{-1}$ collagen gels[46] or in 8 mg mL$^{-1}$ rBM. After gelation, FluoroBrite medium with drug was added: FluoroBrite (Thermo Fisher Scientific) supplemented with GlutaMAX (Thermo Fisher Scientific), 1% Pen/Strep, 10% FBS, 50 ng mL$^{-1}$ EGF, and with vehicle alone (DMSO), 10 µM GM6001, 25 µM GM6001, or 100 µM Marimastat (Tocris). Cells were imaged live in an incubated

chamber (37 °C and 5% $CO_2$) at 20 min intervals with a Leica HCX PL APO ×10/0.40 NA objective overnight.

**Inhibitors.** Inhibitors were implemented in invasive morphology and/or migration assays at concentrations as follows: 10 µM GM6001 (Millipore) or 100 µM marimastat (Tocris) to broadly inhibit MMPs; 70 µM NSC23766 to inhibit Rac1 (Tocris); 10 µM Y-27632 to inhibit ROCK (Sigma); 100 µM CK-666 to inhibit Arp 2/3 (Sigma); 1 µg mL$^{-1}$ monoclonal β1 integrin-blocking antibody (Abcam, P5D2); 50 µM Blebbistatin (Abcam); and 2.5 µM Latrunculin-a (Tocris). Vehicle-alone controls for these inhibitors were as follows: DMSO for GM6001, Marimastat, CK-666, Blebbistatin, and Latrunculin-a; deionized water for NSC23766 and Y-27632; and IgG nonspecific antibody (Sigma, I5381) for β1 integrin-blocking antibody. All inhibitor concentrations followed those used in similar studies: GM6001[6,47], Marimastat[48], NSC23766[15], Y-27632[42], CK-666[49], β1 integrin-blocking antibody[42], Blebbistatin[31,50], and Latrunculin-a[50]. Drug concentrations were also verified in-house using gelatin degradation assays and traction force microscopy experiments (described elsewhere in Methods).

**Immunohistochemistry.** Preparation of gels: For immunohistochemical staining, media were first removed from the gels. The gels were washed once with serum-free DMEM, and then fixed with 4% paraformaldehyde in serum-free DMEM at room temperature for 45–60 min. The gels were then washed three times in PBS containing calcium (cPBS), and then incubated in 30% sucrose in cPBS at 4 °C. The gels were then placed in a mix, which contained 50% of a 30% sucrose in cPBS solution, and the other 50% OCT (Tissue-Tek), for at least 1 day. The media were then removed, the gels were embedded in OCT, and the gels were frozen. The frozen gels were sectioned and stained following standard immunohistochemistry protocols.

Staining sections: The following antibodies and reagents were used for immunohistochemistry: anti-paxillin antibody (1:300; Abcam, Y113), anti-β1 integrin (1:300; Abcam, P5D2), and anti-CTTN (1:300; Abcam, ab33333). Negative controls, where the secondary antibody was added but the primary antibody was not, were conducted to ensure specificity of all stains. Matching secondary antibodies were purchased from Life Technologies. Alexa Fluor 488 phalloidin (Life Technologies, dilutions of 1:50 for β1 integrin/paxillin co-stain and 1:60 for CTTN co-stain) was used to label the actin cytoskeleton, and DAPI (4′,6-diamidino-2-phenylindole) was used to label the nucleus. Fast Green (Sigma), a compound that stains matrix nonspecifically and fluoresces in the near infrared, was used as a nonspecific matrix stain in the adapted Boyden Chamber Invasion assays. The IPN border was identified by thresholding the Fast Green emission signal (red) intensity using ImageJ. ProLong Gold antifade reagent (Life Technologies) was used to minimize photobleaching. Images were acquired using a Leica HC PL APO ×63/1.4 NA oil immersion objective.

**Image analysis.** To quantify morphology of cancer cells, ImageJ was used to manually segment images and to calculate cell circularity, $4\pi \times$ area (perimeter$^{-2}$), whereby 1 indicates a perfect circle, for regions of interest. For migration studies, the centroids of RFP-LifeAct MDA-MB-231 cells were tracked over time using an automated surfaces analysis algorithm in Imaris (Bitplane). Cells that were poorly segmented or present within voids created by dissipated air bubbles were excluded from the analysis. A custom MATLAB script was used to reconstruct cell tracks in 3D and identify cells that displaced greater than one average cell radius for these studies (~14 µm). ImageJ was also used to identify and characterize length and width of protrusions. MATLAB was used to fit protrusion extension distances over time to sinusoids. For matrix densification visualization, ImageJ was used to obtain the signal intensity profile across a cell migration channel, averaged over 10 pixels (cell alive) or 5 pixels (cells lysed) in width. Imaris was used to invert and generate 3D renderings of migration channel z-stacks.

For matrix displacement maps, the procedure to convert bead displacement measurements to matrix displacement fields followed established approaches[31]. While a full 3D matrix displacement analysis would be ideal in order to analyze matrix displacements completely surrounding the cell, this analysis incorporated information from a single z-plane for computational tractability. In brief, cell and bead channel images from a single z-plane were corrected for drift using the ImageJ plugin StackReg. Then, the particle image velocimetry (PIV) ImageJ plugin was used to perform a PIV analysis on the beads, which involved correlating images at time points of interest using a cross-correlation window of 64 pixels. Mesh sizes for this analysis were manually chosen depending on the local bead concentration. This PIV analysis produced a vector field of matrix displacements. Custom MATLAB code was used to display the output as a vector field and as a heat map. In the case of inhibitors with non-motile cells in the HP IPN condition, representative, live cells with stable protrusions, which created observable bead displacement, were used for the analysis. The same rationale was used for analyzing non-motile cells in the MP IPN and LP IPN conditions.

For the HP IPN control condition, 16 vector field maps were included in the average matrix displacement map shown in Fig. 4c. This procedure began by using ImageJ to calculate the cell centroids, based on thresholded fluorescent images of the actin cytoskeleton of the cell. Then, displacement vectors were collected that were within a cutoff radius of ~20 µm away from the cell centroid, as this radius was observed to both capture local matrix deformations and minimize noise due to

other surrounding cells. The cell centroids were translated to a common origin, and then cell outlines and displacement vector fields were rotated such that the cell migration vector was aligned horizontally to the right. Matrix displacement vectors from the 16 cells were binned using a 2D grid, summed within each 10 μm × 10 μm grid space, and the vector magnitudes were divided by the number of cells included in the analysis to create an average vector field.

**Finite element analysis to estimate mechanical stress**. The hydrogels used in this work are viscoelastic, and exhibit creep, or an increase in strain over time, beyond an initial elastic strain under a constant stress (Fig. 1j and Supplementary Fig. 2c). As such, the relationship between stress and strain is time-dependent, and matrix stresses can in general not be calculated from matrix strains using traditional approaches. However, it is noted that the creep of the hydrogels results in larger strains than would be observed in elastic materials with the identical elastic modulus. Therefore, if the hydrogel is assumed to be elastic, stresses calculated using traditional approaches provide an upper bound on the actual stress generated on hydrogel, or a maximum possible stress.

Based on the experimentally obtained matrix displacement maps on single z-planes and the assumption of elasticity, the finite element method was applied to estimate the maximum stress generated on the hydrogels[51]. After determining in-plane strain fields, and making the assumption that strains related to z-axis are negligible, the stress field is calculated using generalized Hooke's law, or

$$\sigma_{ij} = \frac{E}{(1+\nu)}\varepsilon_{ij} + \frac{\nu E}{(1+\nu)(1-2\nu)}\varepsilon_{kk}\delta_{ij}, \quad i,j,k = 1, 2, 3, \tag{3}$$

where $E$ and $\nu$ are elastic modulus and Poisson ratio, respectively. Elastic moduli were determined as the experimentally measured value of ~2 kPa, and the Poisson ratio is chosen as 0.49, as assumed in previous studies[52]. After calculating the stress tensor, the principal stress was calculated and reported to show the maximal stress.

**Statistics**. Statistical analyses were performed using the GraphPad Prism and MATLAB. *P* values provided in figure legends have been corrected for multiple comparisons, where relevant. Additional information about statistical tests performed in these studies has been provided in Supplementary Tables 2 and 3.

## Data availability
All relevant data from this manuscript are available upon request.

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

## Acknowledgements

We acknowledge Ryan Stowers in the Chaudhuri lab for discussions and technical assistance, David Mooney (Harvard University) for helpful discussions, and Marc Levenston (Stanford University) for use of mechanical testing equipment. We also acknowledge the Stanford Cell Sciences Imaging Facility for software access (Imaris) and the Stanford Statistics Department for their consulting services. This work was supported by National Science Foundation Graduate Research Fellowships to K.M.W. and K.A., a Stanford University Vice Provost for Graduate Education Diversifying Academia and Recruiting Excellence (DARE) Fellowship to K.M.W., a Robert and Marvel Kirby Stanford Graduate Fellowship and Stanford ChEM-H Chemistry/Biology Interface Pre-doctoral Training Program to K.A., a National Defense Science and Engineering Graduate Fellowship (NDSEG) to J.C., a Samsung Scholarship to S.N., a Novo Nordisk Foundation Visiting Scholar Fellowship at Stanford Bio-X (NNF15OC0015218) for N.S.R, National Institutes of Health grants (CA205262, GM129098) for L.H. and (K99CA201304) for M.R., and an American Cancer Society grant (RSG-16-028-01) and a National Institutes of Health National Cancer Institute Grant (R37 CA214136) for O.C.

## Author contributions

K.M.W. performed and analyzed most experiments. K.M.W. and O.C designed the experiments and wrote the manuscript. K.A. performed bead tracking experiments and analyzed data on bead displacements. J.C. performed migration experiments in col-1 and RGD-alginate matrices, as well as some creep testing. S.N. performed finite element analyses, J.Y.L. made CTTN-knockdown and TKS5-knockdown cells, and R.D. provided biomaterials assistance. N.S.R. and M.R. provided mouse tumor tissue and R.W. provided human tumor tissue. L.H performed gelatin degradation drug efficacy assays and provided assistance with characterization of invadopodia.

## Additional information

**Competing interests:** The authors declare no competing interests.

