## [Peer Review File · Nature Communications]

Reviewers' Comments:

Reviewer #1:

Remarks to the Author:

General Comments:

The manuscript by Wisdom et al. aims to study the effects of matrix plasticity on cancer cell migration using a synthetic alginate/rBM matrix of interpenetrating networks (IPNs). This experimental system allows the authors to vary the matrix plasticity without affecting the substrate stiffness and ligand density. With this experimental system, the authors found that MDA-MB-231 cells cultured in the high plasticity gels are more protrusive and have higher probability of migration than the ones cultured in low plasticity gels. In addition, they found that cell migration in these synthetic IPNs are protease independent and require the formation of invadopodia, as evident by the immunofluorescent staining of invadopodial markers in cancer cell protrusion. Further experiments showed that this protease-independent migration critically depends on the cells' ability to produce force and deform the matrix. In short, the authors described a mode of cell migration in synthetic IPNs in which matrix deformation through force generated by invadopodia, rather than protease degradation, is the dominate mechanism of migration.

The experiments performed in this study are well thought through, and the analysis was carefully conducted. The authors did an impressive job characterizing the synthetic IPNs and the cellular behaviors inside the IPNs. However, there are two major concerns remaining:

1) The findings presented in this study seem incremental compared to what is already known in the field. As the author outlined in Fig. 5, it is already widely known that cancer cells use amoeboid (protease-independent) and mesenchymal (protease-dependent) modes of migration when navigating through the matrix, and the mode of migration critically depends on ECM architecture and pore size. When the pore size is much smaller than 3 μm , mesenchymal migration, in which cells generate movement by degrading and deforming matrix, dominates. Hence, from what is presented in the manuscript, it seems the authors are describing an extreme form of mesenchymal migration in which the matrix is so malleable that ECM degradation is not needed for migration. In fact, it is possible that the IPNs created a synthetic environment of high ECM plasticity that favors this protease-independent mesenchymal migration. To address this concern, the authors should repeat key experiments in pure Matrigel, a biological relevant ECM, to assess if this new migration mode also exists in system that more closely mimics the in vivo situation.

2) The in vivo relevancy of this finding is not obvious. As pointed out in the last paragraph, the author didn't perform experiments in biologically relevant ECMs to see if this new mode of migration they proposed exists in vivo. Apart from the tumor deformation data presented in Fig. 1C, the author did not carry out any additional in vivo experiments to prove their hypothesis. Although these tumor deformation data are very much appreciated, there is no direct evidence to address whether this protease-independent mesenchymal mode of migration exists inside the tumors. In fact, in the view of this reviewer, an environment in which cells rely solely on force-generated matrix deformation is unlikely to exist in vivo, and may only be associated with a particular set of ECM parameters. The authors should also note that the failure of MMP inhibition as a therapeutic strategy is attributed mostly to off-target effects (Fingleton et al. *Semin Cell Dev Biol*, 2008).

This paper would be significantly strengthened if the authors can show evidence that this new mode of migration can exist in biologically relevant ECM and inside a tumor grown in mice.

Specific Comments:

1) Fig 1C: The pictures of the tumor before and after indentation are not convincing. Maybe a side-

view or a lateral view photo of the tumor before and after indentation showing the compression will be more compelling.

2) Fig 11: Can authors comment on why the partition coefficient for 20 and 30 nm AuNP is different for gels of different plasticity? The authors should also verify the pore size of the IPNs by performing electron microscopy since the measurement of nanoparticle diffusion is a best an indirect method of measuring pore size.

3) The authors mixed rBM with alginate to make IPNs. However, it is not immediately apparent how much rBM the authors used in making the IPNs. This is important information. The authors should provide this information in the main text, rather than in the supplementary table. Does varying the amount of rBM affect the plasticity of IPNs or cell migration results?

4) Supplementary Fig. 6E: MDA231 cells have the ability to degrade basement membrane, since they secrete MMP2 and MMP9. The authors showed in Fig. S6E that when the basement membrane is removed from the IPNs and replaced with RGD ligands, the migratory behaviors of the cancer cells was inhibited. This seems to point to the fact that rBM is an important component of IPNs that allows the cell to migrate, which raises the possibility that protease-dependent migration still plays a role in the system (i.e. these data seem to contradict the GM6001 data in Fig. 2). The authors' explanation of rBM ligands acting as activators of migration is possible, but this claim would be strengthened by measuring the migration of cancer cells in RGD-functionalized IPNs treated with growth factors that induce migration.

5) As the authors noted in the manuscript (page 9), 4T1 cells and MCF7 cells do not form protrusions in the HP IPNs compared to MDA231 and HT1080 cells, indicating the ability of the invadopodia to act through degradation and force may be cell dependent. To expand upon this result, the authors could compare the degree of force generation by 4T1 cells and MDA231 cells to assess if MDA231 cells can generate more force. Similarly, the ability of 4T1 and MDA231 cells to degrade basement membrane should be accessed, to see if 4T1 cells have enhanced ability to degrade rBM compared to MDA231 cells.

6) To prove that cancer cells permanently deform the IPNs (page 8), the authors could decellularize the IPNs (using Triton X) and assess if the channels formed by the cells remain after the cells are removed from the matrix.

7) The authors' arguments for the reason behind the similar migration speed between cancer cells seeded in HP (high plasticity) IPNs and LP (low plasticity) IPNs are not convincing. If HP IPNs are easier to deform than LP IPNs, the cells should form channels faster or with higher frequency, thus allowing cells to migrate faster. Perhaps a better explanation is that in the IPNs, the machinery that governs the speed of migration is decoupled from the deformation of the matrix. The authors should stimulate cells with EGF to see if treating cells with growth factors that induce migration can allow cells to migrate faster.

8) On page 10, there is a typo in the sentence (highlighted): "The elevated viscosity of breast tumor tissue that has been observed clinically¹⁵, the aberrant crosslinking and matrix architectures associated with tumor progression¹⁶, and our data revealing substantial mechanical plasticity of human and mouse tumor tissue all point towards the likelihood of both the BM and stromal matrix exhibiting some degree of mechanical plasticity during breast cancer."

9) In the discussion, the authors should note that although the density of the matrix often increases during breast cancer progression, the matrix is often aligned, providing large gaps and fiber bundles that greatly enhance cancer cell migration (Conklin et al., *American Journal of Pathology*, 2011), (PP Provenzano, *BMC Medicine*, 2006), (Patsialou et al., *Intravital*, 2013), and (Gligorijevic et al. *PLOS Biol*, 2014).

10) In Fig. S6, the authors showed that GM6001 treatment on cancer cells cultured in 4.0 mg/mL rat tail collagen had no effect on cell migration speed, displacement, and the probability of cell migration. However, numerous studies have shown that treating cells with GM6001 in rat tail collagen, even at collagen density as low as 1.7 mg/mL, can lead to inhibition of cancer cell migration (Wolf et al., *JCB*, 2013), (Fisher et al., *Mol Cancer*, 2006), (Sabeh et al, *JCB*, 2009), and (Carey et al, *Scientific Reports*, 2017). How can the authors rationalized this discrepancy?

11) In Fig. S8, can authors comment on why inhibiting beta 1 integrin, ROCK, and Arp2/3 did not affect migration speed and displacement? It is interesting that in the IPN matrix, cell migration probability seems to be decoupled from migration speed/displacement. Is the same phenomenon

observed in biologically-relevant ECM like pure rBM or collagen I matrix?

Reviewer #2:

Remarks to the Author:

This manuscript is potentially interesting, as it shows that a property that it is conceptually different to stiffness and viscoelasticity might play a role in cell migration. If the authors were to robustly and convincingly demonstrate that the cells were using forces to exceed the elastic limit of the gels and permanently deform them, then I would think that this is a novel and interesting addition to what we already know about cells migrating in 3D. However, introducing a new property has to be done with care as plasticity involves permanent deformation of a material beyond the elastic limit. I am not sure this is really what happens here as the authors have not robustly demonstrated mechanically that their alginate gels respond with true plasticity.

The characterisation of the IPN is rather poor. They have performed a very simple creep analysis (fig 1i, 1j) - and one more at 150 Pa in after the revision. This allows measuring the permanent deformation ('plasticity') after removing force on the material. The control indentations show that there are permanent deformations in the long term after removing the force. However, it has not been convincingly shown that cells apply forces on the material that are high enough to result in permanent deformations. This is not straightforward from the material characterisation shown in Figure 1, e.g. using stress-strain curves. They need to more robustly demonstrate that their system can be deformed permanently by applying force and that this is what the cells are doing.

The system they use is rather obscure: IPN made of reconstituted BM and alginate which does not work if the BM is substituted by alginate (the one which provides plasticity) functionalised with RGD. What's the origin of this 'plasticity' of the material at the molecular scale? Are their alginate-RGD hydrogels 'plastic'? If they are, then it is the combined effect of the BM and alginate that allows migration but the role of the BM in the overall process has not been addressed at all?

Reviewer #3:

Remarks to the Author:

In this resubmitted manuscript by Wisdom et. al., entitled "Matrix mechanical plasticity regulates cancer cell migration through confining microenvironments", researchers from the Chaudhuri group build on their earlier conjecture that protease-independent cell migration occurs in overly-confining ECM as a function of mechanical remodeling of the matrix. The manuscript has been strengthened overall by a refocusing of the terminology onto plasticity, a more rigorous protease inhibitor assay panel, and more clarifying discussion. This reviewer supports publication after two minor points are addressed:

1. The reviewer apologizes for referring to line 565 as a typo when in reality it was a grammatical error. "Chemoattractant medium was replaced, and (the) plate was gently shaken on an orbital shaker".

2. It appears that Reviewer 1 and Reviewer 2 also were confused about the results in Extended Figure 4, meaning that it is highly likely that the scientific audience of this article will also be confused. The author's response to all three reviewers raising this concern was, essentially:

"The mechanism of migration we observe involves repetitive extension of invadopodial protrusions over the span of many hours, which opens up a channel, followed by cells migrating by then squeezing through the opened channel. The data are consistent with the explanation that what dictates the speed and distance is the length of the channel formed by the invadopodial protrusion. Under this explanation, the cells do not migrate until a channel of a sufficient width and distance is

formed, and then, once such a channel is formed, migrate the length of that channel. Therefore, the migration speeds and distances observed in the different conditions would necessarily be similar."

This reviewer does not doubt the conclusions of the authors, but feels strongly that a clearer answer to this question must be provided. The explanation given states that the formation of the 'channel' is the rate limiting step to the migration, and thus, the discrepancy between percentage of motile cells in HP and LP matrices (11.5% vs. 2%) is due to a six-fold increase in the ability of cells to form pore-like 'channels' in the matrix. Then, once the pore has been formed, migration occurs with identical mechanisms in both HP and LP matrices, leading to identical values for average speed, maximum speed, and displacement. The discussion attempts to summarize the point:

"Under this explanation, the cells do not migrate until a channel of a sufficient width and distance is formed, and then, once such a channel is formed, migrate the length of that channel."

What remains unclear is a) how long is the distance of the pore, and b) what happens after these pores have been formed? Is the cell limited by the distance of the pore that is formed? In Fig. 4h, the pore only appears to be 2-4 μm long. In addition, the cell that has traversed this pore, which took many hours to build in the first place, was able to deform the matrix to the right of the pore to a degree equal to that on the left side, where the cell was initially encapsulated and thus the stress would have likely been much smaller. Furthermore, no densification seems apparent on the right side of Fig. 4h, despite a far higher degree of deformation that would have been present in the pore region.

The only change that this reviewer believes is necessary to clear up this confusion is a more detailed cartoon diagram in figure 5 which hypothesizes exactly how the cell goes from 'New Mode' panel 2 to 'New Mode' panel 3. An additional panel in between these two time points showing the intermediate step would be very helpful. Does the entire cell body flow from the left side to the right side, to the limit of the furthest point of the invadopodia, then expand isotropically until we reach the 'New Mode' panel 3? Or did the invadopodia plastically displace the ECM to such a high degree that there is already a void in the matrix prior to cell translocation?

To summarize, this reviewer is not requesting any new experiments and believes that the manuscript is suitable for publication, but believes it is in the best interest of the authors to clarify their proposed 'new mode' in a way that is in agreement with the data in Extended Figure 4 and the images in Figure 4 g-h.

Reviewer #4:

Remarks to the Author:

The authors have done a great job in addressing all of my previous concerns. They have added significant additional experiments and analyses to support their conclusions. I feel that this manuscript should be published in its current form.

RESPONSE TO REVIEWERS

We thank the reviewers for another thorough review of the manuscript, and for their constructive critiques and suggestions. To address the referees' remaining comments, we have performed additional sets of experiments and revised the manuscript text accordingly. We believe that this revision has enhanced the clarity and completeness of the manuscript. A specific point-by-point response follows, and key changes to the manuscript text have been underlined.

Response to Reviewer #1

General Comments:

The manuscript by Wisdom et al. aims to study the effects of matrix plasticity on cancer cell migration using a synthetic alginate/rBM matrix of interpenetrating networks (IPNs). This experimental system allows the authors to vary the matrix plasticity without affecting the substrate stiffness and ligand density. With this experimental system, the authors found that MDA-MB-231 cells cultured in the high plasticity gels are more protrusive and have higher probability of migration than the ones cultured in low plasticity gels. In addition, they found that cell migration in these synthetic IPNs are protease independent and require the formation of invadopodia, as evident by the immunofluorescent staining of invadopodial markers in cancer cell protrusion. Further experiments showed that this protease-independent migration critically depends on the cells' ability to produce force and deform the matrix. In short, the authors described a mode of cell migration in synthetic IPNs in which matrix deformation through force generated by invadopodia, rather than protease degradation, is the dominant mechanism of migration. The experiments performed in this study are well thought through, and the analysis was carefully conducted. The authors did an impressive job characterizing the synthetic IPNs and the cellular behaviors inside the IPNs. However, there are two major concerns remaining:

We thank this reviewer for reviewing the manuscript a second time. We were pleased that we were able to address many of the concerns raised by this reviewer in the initial round of review. We are also glad that this reviewer finds our experiments to be “well thought through” and “carefully conducted”, and that this reviewer was impressed by our characterization of the IPNs and cellular behavior inside the IPNs. In response to this reviewer's critique, we have characterized the mechanical plasticity of biological ECMs, analyzed the role of proteases in cell migration in biological ECMs, and conducted a more thorough analysis of the channels formed by migrating cells in high plasticity IPNs.

Major concerns

1) The findings presented in this study seem incremental compared to what is already known in the field. As the author outline in Fig. 5, it is already widely known that cancer cells use amoeboid (protease-independent) and mesenchymal (protease-dependent) modes of migration when navigating through the matrix, and the mode of migration critically depends on ECM architecture and pore size. When the pore size is much smaller than 3 μm , mesenchymal migration, in which cells generate movement by degrading and deforming matrix, dominates. Hence, from what is presented in the

manuscript, it seems the authors are describing an extreme form of mesenchymal migration in which the matrix is so malleable that ECM degradation is not needed for migration. In fact, it is possible that the IPNs created a synthetic environment of high ECM plasticity that favors this protease-independent mesenchymal migration. To address this concern, the authors should repeat key experiments in pure Matrigel, a biological relevant ECM, to assess if this new migration mode also exists in system that more closely mimics the in vivo situation.

As the reviewer notes, the current understanding of mesenchymal migration is that it is protease-dependent and requires matrix degradation. We agree with the reviewer that the similar machinery used in protease-dependent migration and plasticity-mediated migration (e.g. invadopodia) could represent two extremes of mesenchymal migration. We have noted this excellent idea in the discussion section of the manuscript. However, we note that the current thinking is that proteases are an absolute requirement for migration through matrices with nanometer-sized pores, through which the nucleus cannot be squeezed. For example, the recent review by Peter Friedl and colleagues notes: “*Nearly impenetrable, dense ECM impedes cell migration and requires particular abilities, such as the capacity to strongly deform the nucleus and/or to proteolytically degrade ECM and generate space¹¹¹. Collagen-rich stroma and the basement membrane are examples of such high-density environments^{116,158}” (Friedl et al., *Nature Cell Biology*, 2018). Thus, our finding that matrix plasticity can also mediate migration through dense ECM introduces the significant conceptual advance that proteases are not required for migration through nanoporous matrix if the matrix exhibits sufficient mechanical plasticity.*

We thank the reviewer for the suggestion of repeating key experiments with pure Matrigel. First, we performed additional creep and recovery tests on Matrigel-only hydrogels and collagen-1-only hydrogels in order to establish the mechanical plasticity of these commonly used, biologically relevant ECMs. Importantly, we find that Matrigel and collagen hydrogels are actually even more mechanically plastic than our high plasticity IPNs (Figure R1 / Fig. 1k; Matrigel is indicated as rBM). While these reconstituted ECMs may not fully capture the architecture and crosslinking of biological ECMs *in vivo*, these measurements do further point toward the biological relevance of mechanical plasticity in ECM.

Figure R1: Plasticity measurements from a 10 Pa Creep and Recovery Test

After establishing the mechanical plasticity of Matrigel, we then performed additional cell migration studies in 8 mg/mL Matrigel and assessed the role of proteases. We found that the addition of broad spectrum protease inhibitor GM6001 does not appreciably change the probability of cell migration, motile cell displacement length, or motile cell mean migration speed (Figure R2 / Supplementary Fig. 6c). However, there is a significant, though small, reduction in the maximum cell speed. Nonetheless, these findings indicate that proteases are dispensable for cancer cells migrating through Matrigel, which exhibits high mechanical plasticity. This is consistent with other studies of cell migration in Matrigel (Poincloux *et al.*, *PNAS*, 2011; Rowe and Weiss, *Annu. Rev. Cell Dev. Biol.*, 2009; Sodek *et al.*, *BMC Cancer*, 2008).

Figure R2: rBM Migration Assay

We agree with this reviewer that Matrigel contains proteins that are biologically relevant, but note that there are some distinctions with *in vivo* BM. The loosely connected network of weakly interacting proteins in Matrigel fails to fully mimic the architecture and stiffness of basement membrane *in vivo* (Hotary *et al.*, 2006; Kleinman and Martin, *Semin. Cancer Biol.*, 2005; Rowe and Weiss, *Annu. Rev. Cell Dev. Biol.*, 2009; Sabeh *et al.*, *J. Cell Biol.*, 2009). Our inclusion of alginate, a polymer inert to cells, alongside the Matrigel component enables us to not only stiffen the combined network so that it is

comparable to tumor tissue (Figure R3 / Supplementary Fig. 2c), but also to tune mechanical plasticity across the series of IPN hydrogels.

Figure R3: Stiffness of HP IPN and pure rBM

2) *The in vivo relevancy of this finding is not obvious. As pointed out in the last paragraph, the author didn't perform experiments in biologically relevant ECMs to see if this new mode of migration they proposed exists in vivo. Apart from the tumor deformation data presented in Fig. 1C, the author did not carry out any additional in vivo experiments to prove their hypothesis. Although these tumor deformation data are very much appreciated, there is no direct evidence to address whether this protease-independent mesenchymal mode of migration exists inside the tumors. In fact, in the view of this reviewer, an environment in which cells rely solely on force-generated matrix deformation is unlikely to exist in vivo, and may only be associated with a particular set of ECM parameters. The authors should also note that the failure of MMP inhibition as a therapeutic strategy is attributed mostly to off-target effects (Fingleton et al. Semin Cell Dev Biol, 2008). This paper would be significantly strengthened if the authors can show evidence that this new mode of migration can exist in biologically relevant ECM and inside a tumor grown in mice.*

As discussed in response to the previous point, we have found protease-independent migration in the biologically relevant ECM Matrigel, which exhibits high mechanical plasticity. We have also added a comment into the manuscript acknowledging this reviewer's important point regarding off-target effects contributing to failure of MMP inhibition as a therapeutic strategy. While it would be ideal to support our findings with *in vivo* mouse tumor data, there is not a good mouse model of initial invasion of basement membrane (i.e. out of the primary tumor). Furthermore, it would be challenging to assess the independent contributions of proteases versus mechanical plasticity in a tissue model, and therefore generate unambiguous conclusions.

Specific Comments:

1) *Fig 1C: The pictures of the tumor before and after indentation are not convincing. Maybe a side-view or a lateral view photo of the tumor before and after indentation showing the compression will be more compelling.*

We agree with this reviewer that the indentation of the tissue surface is difficult to see. While multiple angles were explored, imaging wet, shiny tissue samples was challenging. However, the permanent shifting in the location of discolored regions of tissue is unambiguous evidence of mechanical plasticity that can be gleaned from this figure. In the manuscript, we now more clearly emphasize the quantitative data from the mechanical testing and the shifted location of the discolored regions of tissue as evidence of mechanical plasticity.

2) *Fig 11: Can authors comment on why the partition coefficient for 20 and 30 nm AuNP is different for gels of different plasticity? The authors should also verify the pore size of the IPNs by performing electron microscopy since the measurement of nanoparticle diffusion is a best an indirect method of measuring pore size.*

We thank the reviewer for the suggestions. First, we emphasize that our goal with the nanoparticle studies was to put an upper bound on the pore size, as opposed to providing a precise measurement of the pore size distribution. The difference in the partition coefficient of the 20 and 30 nm AuNP diffusion could be due to the weak negative surface charge of the gold nanoparticles, as evidenced by the zeta potential provided by the manufacturer. Given that our IPNs are crosslinked with different amounts of positively charged calcium ion, the gold nanoparticles may be somewhat more attracted to the alginate networks crosslinked with higher concentrations of calcium.

While electron microscopy provides striking images, it may not accurately capture the matrix architecture. This is because SEM sample preparation requires freeze-drying. It has been shown that “*rapid collapse of the hydrogel structure is seen upon removal of the bulk and interfacial water in the frozen state*” during SEM sample preparation of hydrogels (Miller and Peppas, *Biomaterials*, 1986). While recent electron microscopy advancements can enable imaging of hydrated specimens, the spatial resolution (tens of microns) would not be sufficient to characterize the pore size of our nanoporous hydrogels (Denisin and Pruitt, *ACS Appl. Mater. Interfaces*, 2016; Joubert, 2017).

3) *The authors mixed rBM with alginate to make IPNs. However, it is not immediately apparent how much rBM the authors used in making the IPNs. This is important information. The authors should provide this information in the main text, rather than in the supplementary table. Does varying the amount of rBM affect the plasticity of IPNs or cell migration results.*

We have now clarified the rBM concentration of these IPNs in the main body of the manuscript. All IPNs contain 4.4 mg/mL rBM, which was previously found to provide a sufficient biological stimulus for MCF10A mammary epithelial cells to form acinar structures (Chaudhuri *et al.*, *Nat. Mater.*, 2014).

We addressed this reviewer’s second question, regarding the contribution of rBM to the plasticity of the IPNs, by performing new creep and recovery tests the low molecular weight (LMW) alginate network of the HP IPN alone (i.e. with no rBM at all). We find that the mechanical plasticity of the alginate network alone and of the complete HP IPN are comparable (Figure R4 / Fig. 1k). We were unable to increase the

concentration of rBM in the IPN hydrogels substantially beyond 4.4 mg/mL: the solubility limit of alginate and the concentration of rBM matrix supplied commercially prevent us technically from forming IPNs with higher concentrations of rBM. While it would be interesting to further study the impact of varying rBM on cell migration between 0 and 4.4 mg/mL, such studies would be tangential to the main focus of this manuscript.

Figure R4: Plasticity measurements from a 10 Pa Creep and Recovery Test

4) *Supplementary Fig. 6E: MDA231 cells have the ability to degrade basement membrane, since they secrete MMP2 and MMP9. The authors showed in Fig. S6E that when the basement membrane is removed from the IPNs and replaced with RGD ligands, the migratory behaviors of the cancer cells was inhibited. This seems to point to the fact that rBM is an important component of IPNs that allows the cell to migrate, which raises the possibility that protease-dependent migration still plays a role in the system (i.e. these data seem to contradict the GM6001 data in Fig. 2). The authors' explanation of rBM ligands acting as activators of migration is possible, but this claim would be strengthened by measuring the migration of cancer cells in RGD-functionalized IPNs treated with growth factors that induce migration.*

The role of rBM ligands in general, and of laminins in particular, in activating migration and invasion is well known (Chen *et al.*, 2016; Kruegel and Miosge, *Cell. Mol. Life Sci.*, 2010; Marinkovich, *Nat. Rev. Cancer*, 2007; Yurchenco *et al.*, *Cold Spring Harb. Perspect. Biol.*, 2011). With regard to this reviewer's suggestion, we note that all migration studies presented in this manuscript have already been performed using medium supplemented with 50 ng/mL Epidermal Growth Factor (EGF), as is standard in invasion assays (Chen *et al.*, 2016; Moshfegh *et al.*, *Nat. Cell Biol.*, 2014; Wang *et al.*, *Exp. Cell Res.*, 2004; Zervantonakis *et al.*, *PNAS*, 2012). We have now clarified this experimental detail in the manuscript.

5) *As the authors noted in the manuscript (page 9), 4T1 cells and MCF7 cells do not form protrusions in the HP IPNs compared to MDA231 and HT1080 cells, indicating the ability of the invadopodia to act through degradation and force may be cell dependent. To expand upon this result, the authors could compare the degree of force generation by 4T1 cells and MDA231 cells to assess if MDA231 cells can generate more force.*

Similarly, the ability of 4T1 and MDA231 cells to degrade basement membrane should be accessed, to see if 4T1 cells have enhanced ability to degrade rBM compared to MDA231 cells.

We agree that a broad survey of cancer cell line-specific differences in abilities for protease degradation mediated migration and cell-generated force mediated migration would be very interesting. However, such a study is beyond the scope of this manuscript, which is focused on elucidating the mechanism underlying the force-generation mediated mode of migration.

6) To prove that cancer cells permanently deform the IPNs (page 8), the authors could decellularize the IPNs (using Triton X) and assess if the channels formed by the cells remain after the cells are removed from the matrix.

We thank the reviewer for this excellent idea. To strengthen the claim that the cancer cells permanently deform the IPNs, we have performed an additional experiment with cells encapsulated in HP IPNs made with fluorescent alginate, but here, 1% Triton-X and 50 μ M Cytochalasin D were added to lyse the cells and depolymerize actin networks, respectively. We observed that channels remain 3-4 hours after cells have been removed from the matrix. We have now added these data into the manuscript (Figure R5 / Fig. 4h,i), (Figure R6 / Supplementary Fig. 9f,g).

Figure R5: Permanent channels in HP IPN hydrogels. a, Fluorescent alginate intensity across cell migration channel (line from A to B). b, Similarly, fluorescent alginate intensity across cell migration channels, 3-4 hours after cell lysis and actin network depolymerization. Scale bars are 10 μ m.

Figure R6: Very long, permanent channel in HP IPN. a, A lasting migration channel that spans several z-planes. Images taken 2.4 μm apart as part of a z-stack. Scale bar is 10 μm . b, Signal was inverted and the channel was rendered in 3D.

7) *The authors' arguments for the reason behind the similar migration speed between cancer cells seeded in HP (high plasticity) IPNs and LP (low plasticity) IPNs are not convincing. If HP IPNs are easier to deform than LP IPNs, the cells should form channels faster or with higher frequency, thus allowing cells to migrate faster. Perhaps a better explanation is that in the IPNs, the machinery that governs the speed of migration is decoupled from the deformation of the matrix. The authors should stimulate cells with EGF to see if treating cells with growth factors that induce migration can allow cells to migrate faster.*

We thank the reviewer for suggesting this word choice, which succinctly captures how we interpret the findings as well. We have added a comment regarding the decoupling of the cell making the channel, and the cell migrating through that channel, in the manuscript. Regarding the second point raised, we note that all studies in this manuscript using cancer cells have been performed with 50 ng/mL EGF added to the medium. We have emphasized this experimental detail in the manuscript.

8) *On page 10, there is a typo in the sentence (highlighted): "The elevated viscosity of breast tumor tissue that has been observed clinically¹⁵, the aberrant crosslinking and matrix architectures associated with tumor progression¹⁶, and our data revealing substantial mechanical plasticity of human and mouse tumor tissue all point towards the likelihood of both the BM and stromal matrix exhibiting some degree of mechanical plasticity during breast cancer."*

Unfortunately the highlighted formatting did not transfer as part of this review process. If the suggestion was perhaps to change "towards" to "toward", we have done this in the revised manuscript.

9) *In the discussion, the authors should note that although the density of the matrix often increases during breast cancer progression, the matrix is often aligned, providing large gaps and fiber bundles that greatly enhance cancer cell migration (Conklin et al., American Journal of Pathology, 2011), (PP Provenzano, BMC Medicine, 2006), (Patsialou et al., Intravital, 2013), and (Gligorijevic et al. PLOS Biol, 2014).*

We appreciate this reviewer's suggestion and have added this point into our discussion.

10) *In Fig. S6, the authors showed that GM6001 treatment on cancer cells cultured in 4.0 mg/mL rat tail collagen had no effect on cell migration speed, displacement, and the probability of cell migration. However, numerous studies have shown that treating cells with GM6001 in rat tail collagen, even at collagen density as low as 1.7 mg/mL, can lead to inhibition of cancer cell migration (Wolf et al., JCB, 2013), (Fisher et al., Mol Cancer, 2006), (Sabeh et al, JCB, 2009), and (Carey et al, Scientific Reports, 2017). How can the authors rationalized this discrepancy?*

We appreciate the reviewer's request to clarify these discrepancies. Firstly, we note that two of the studies mentioned utilized rat tail collagen with intact telopeptides, which allows for extensive covalent crosslinking of the collagen gels (Sabeh *et al.*, *J. Cell Biol.*, 2009; Wolf *et al.*, *J. Cell Biol.*, 2013). In the control studies in our manuscript, we used the commercially available rat tail collagen-1 product, which does not contain these telopeptides. Sabeh *et al.* and others report that dense covalent crosslinks prevent MMP-independent migration, which explains discrepancies between assays using different collagen products (Sabeh *et al.*, *J. Cell Biol.*, 2009; Sodek *et al.*, *BMC Cancer*, 2008). We note that increased covalent crosslinking is expected to decrease mechanical plasticity of the matrices (Nam *et al.*, *Biophys. J.*, 2016; *PNAS*, 2016). Furthermore, in the 2013 JCB paper, Wolf *et al.* note that they used a mutant MDA-MB-231 cell line that artificially expresses constitutively active MT1-MMP, which has been shown to increase tumor cell invasion in 3D collagen matrices (Moss *et al.*, *J. Biol. Chem.*, 2009). This genetic manipulation likely increases cell sensitivity to protease inhibition, particularly in an environment with intact telopeptides. By contrast, our studies have been conducted using MDA-MB-231 cells transfected only to express fluorescent actin and using collagen-1 without telopeptides. We have added clarifying comments with respect to these differences into the manuscript.

The other two studies brought up by this reviewer were conducted using MCF10A cells (Carey *et al.*, *Am. J. Physiol. - Cell Physiol.*, 2015) and HT-1080 cells (Fisher *et al.*, *Mol. Cancer*, 2006), whereas our study focused on MDA-MB-231 cells. We have emphasized the parts of the manuscript that include caveats regarding the degree to which our findings may be cell type-dependent.

11) *In Fig. S8, can authors comment on why inhibiting beta 1 integrin, ROCK, and Arp2/3 did not affect migration speed and displacement? It is interesting that in the IPN matrix, cell migration probability seems to be decoupled from migration speed/displacement. Is the same phenomenon observed in biologically-relevant ECM like pure rBM or collagen I matrix?*

We agree that it is an interesting finding that cell migration probability seems to be decoupled from migration speed/displacement. This could explain why inhibiting $\beta 1$ integrin and Arp 2/3 significantly impacted cell morphology and cell migration probability, but not migration speed and displacement (Fig. 4a, Supplementary Fig. 7f, Supplementary Fig. 8). We have added a point to this regard in the manuscript. While it would be interesting to examine this question in pure rBM or collagen-1, this would be beyond the scope of this manuscript, as these matrices are known to be conducive to a variety of different migration modes (Friedl and Wolf, *J. Cell Biol.*, 2010).

Response to Reviewer #2

General Comments:

This manuscript is potentially interesting, as it shows that a property that it is conceptually different to stiffness and viscoelasticity might play a role in cell migration. If the authors were to robustly and convincingly demonstrate that the cells were using forces to exceed the elastic limit of the gels and permanently deform them, then I would think that this is a novel and interesting addition to what we already know about cells migrating in 3D. However, introducing a new property has to be done with care as plasticity involves permanent deformation of a material beyond the elastic limit. I am not sure this is really what happens here as the authors have not robustly demonstrated mechanically that their alginate gels respond with true plasticity.

We are pleased that this reviewer believes that our studies offer a “novel and interesting addition to what we already know about cells migrating in 3D”. The key concern raised by this reviewer is of whether cells are applying a force that is sufficient to deform the material beyond its elastic limit. We have now conducted further characterization of plasticity. We find that the HP IPNs exhibit substantial mechanical plasticity even at stresses as low as 10 Pa and provide further evidence that the IPNs are being plastically deformed by the cells.

The characterisation of the IPN is rather poor. They have performed a very simple creep analysis (fig 1i, 1j) - and one more at 150 Pa in after the revision. This allows measuring the permanent deformation ('plasticity') after removing force on the material. The control indentations show that there are permanent deformations in the long term after removing the force. However, it has not been convincingly shown that cells apply forces on the material that are high enough to result in permanent deformations. This is not straightforward from the material characterisation shown in Figure 1, e.g. using stress-strain curves. They need to more robustly demonstrate that their system can be deformed permanently by applying force and that this is what the cells are doing.

We note that the creep and recovery tests like those we have performed are standard to measure the degree of permanent deformation of a material. We reference several key textbooks to support this, including p. 2-5 (Findley *et al.*, *Dover Publ.*, 1976), p. 111-112 (Shames and Cozzarelli, 1997), and p. 54-66 (Ward and Hadley, 2004), as well as a recent article measuring viscoplasticity in cells (Bonakdar *et al.*, *Nat. Mater.*, 2016). We are not aware of other plasticity tests typically used in the biomaterials field to quantitatively characterize mechanical plasticity.

More broadly, the reviewer raises the important point of whether cells are indeed applying a force sufficient to induce permanent deformations in the matrices. To further address this reviewer's concern, we have performed additional creep and recovery tests of the high plasticity IPNs at much lower stresses (10 Pa and 50 Pa). These mechanical testing data indicate that even a stress of 10 Pa is sufficient to induce plastic deformation of the HP IPN (permanent deformation of ~ 20% for 10 Pa of stress applied for 1 hour), demonstrating that the elastic-plastic yield stress of the HP IPN hydrogel is below 10 Pa (Figure R7 / Supplementary Fig. 2e). A stress of 10 Pa is well below those which cells

typically exert on the extracellular matrix: cells can exert protrusive stresses between 2-10 kPa using 2D lamellipodia (Prass *et al.*, *J. Cell Biol.*, 2006) and traction stresses from 1-5 kPa through cell-ECM adhesions in 3D culture (Franck *et al.*, *PLoS One*, 2011; Legant *et al.*, *Nat. Methods*, 2010). Given these data, we would expect cells to exert stresses above 10 Pa, and thereby permanently deform the matrix. We note that due to noise and instrument resolution, we were unable to robustly apply stresses below 10 Pa on these materials.

Figure R7: Plasticity vs. Creep Stress

To further strengthen the claim that the cancer cells permanently deform the IPNs, we have performed an additional experiment with cells encapsulated in HP IPNs made with fluorescent alginate, but here, 1% Triton-X and 50 μ M Cytochalasin D were added to lyse the cells and depolymerize actin networks, respectively. We observed that channels remain 3-4 hours after cells have been removed from the matrix. We have now added these data into the manuscript (Figure R8 / Fig. 4h,i),(Figure R9 / Supplementary Fig. 9f,g). Combined, these mechanical testing and imaging data provide independent support for the idea that cells are capable of using force to permanently deform the high plasticity hydrogel.

Figure R8: Permanent channels left in HP IPNs. a, Fluorescent alginate intensity across cell migration channel (line from A to B). b, Similarly, fluorescent alginate intensity across cell migration channels, 3-4 hours after cell lysis and actin network depolymerization. Scale bars are 10 μm .

Figure R9: Very long, permanent channel left in HP IPN. a, A lasting migration channel that spans several z-planes. Images taken 2.4 μm apart as part of a z-stack. Scale bar is 10 μm . b, Signal was inverted and the channel was rendered in 3D.

The system they use is rather obscure: IPN made of reconstituted BM and alginate which does not work if the BM is substituted by alginate (the one which provides plasticity) functionalised with RGD. What's the origin of this 'plasticity' of the material at the molecular scale? Are their alginate-RGD hydrogels 'plastic'? If they are, then it is the combined effect of the BM and alginate that allows migration but the role of the BM in the overall process has not been addressed at all?

We thank the reviewer for this question. We reiterate that the design of our materials system was motivated by the goals to i) utilize biological ligands present in basement membrane, ii) mimic the matrix nanoporosity and stiffness that cancer cells encounter during initial invasion of the BM by a primary tumor, and iii) enable tunable mechanical plasticity of the ECM, while keeping constant ligand density and stiffness. Both rBM and alginate are commonly used for 3D cell culture (Caliari and Burdick, *Nat. Methods*, 2016).

In order to determine the mechanical contributions of the rBM and alginate networks, we conducted additional mechanical tests. We have now performed creep and

recovery testing on pure rBM, the alginate component of the HP IPN alone, and the entire HP IPN. These additional tests were performed at a stress of 10 Pa. We find that while pure rBM exhibits high plasticity (75%), the alginate component of the network alone exhibits plasticity that is comparable to the combined alginate and rBM networks in the HP IPN (~ 20%) (Figure R10 / Fig. 1k). We note that conjugation of the alginate with RGD is not expected to alter the mechanical properties of the alginate network substantially. Importantly, all IPNs used for these studies exhibit a stiffness on the order of tumor tissue, which is an order of magnitude stiffer than rBM alone, thanks to the interpenetrating alginate networks (Figure R11 / Supplementary Fig. 2c).

The origins of plasticity in alginate, rBM, and the IPNs ultimately arises from weak bonds that can unbind and rebind. When mechanical stresses are applied to the alginate hydrogels, the ionic crosslinkers can unbind allowing polymer flow, and then can rebind again leading to plastic deformations in the hydrogel (Zhao *et al.*, *J. Appl. Phys.*, 2010). A note regarding this point has been added to the second paragraph of the manuscript text. High plasticity of the rBM, which has been noted previously (Nam *et al.*, *Biophys. J.*, 2016), likely arises out of the weak, nonspecific protein binding in rBM (Kleinman and Martin, *Semin. Cancer Biol.*, 2005).

Biologically, the rBM component of these IPNs is rich in signaling molecules, such as laminins in particular, that stimulate migration and invasion (Chen *et al.*, 2016; Kruegel and Miosge, *Cell. Mol. Life Sci.*, 2010; Marinkovich, *Nat. Rev. Cancer*, 2007; Yurchenco *et al.*, *Cold Spring Harb. Perspect. Biol.*, 2011). We have more clearly emphasized the mechanical and biological contributions of the rBM and alginate networks to the IPNs in the revised manuscript.

Figure R10: Plasticity measurements from a 10 Pa Creep and Recovery Test

Figure R11: Stiffness of HP IPN and rBM

Response to Reviewer #3

General Comments:

In this resubmitted manuscript by Wisdom et. al., entitled “Matrix mechanical plasticity regulates cancer cell migration through confining microenvironments”, researchers from the Chaudhuri group build on their earlier conjecture that protease-independent cell migration occurs in overly-confining ECM as a function of mechanical remodeling of the matrix. The manuscript has been strengthened overall by a refocusing of the terminology

onto plasticity, a more rigorous protease inhibitor assay panel, and more clarifying discussion. This reviewer supports publication after two minor points are addressed.

We thank this reviewer for a second review of the work. We are pleased that this reviewer supports publication. We address the minor points this reviewer raised below.

Specific Comments

1) *The reviewer apologizes for referring to line 565 as a typo when in reality it was a grammatical error. “Chemoattractant medium was replaced, and (the) plate was gently shaken on an orbital shaker”.*

We thank the reviewer for pointing this out, the grammatical error has been fixed.

2) *It appears that Reviewer 1 and Reviewer 2 also were confused about the results in Extended Figure 4, meaning that it is highly likely that the scientific audience of this article will also be confused. The author’s response to all three reviewers raising this concern was, essentially: “The mechanism of migration we observe involves repetitive extension of invadopodial protrusions over the span of many hours, which opens up a channel, followed by cells migrating by then squeezing through the opened channel. The data are consistent with the explanation that what dictates the speed and distance is the length of the channel formed by the invadopodial protrusion. Under this explanation, the cells do not migrate until a channel of a sufficient width and distance is formed, and then, once such a channel is formed, migrate the length of that channel. Therefore, the migration speeds and distances observed in the different conditions would necessarily be similar.”*

This reviewer does not doubt the conclusions of the authors, but feels strongly that a clearer answer to this question must be provided. The explanation given states that the formation of the ‘channel’ is the rate limiting step to the migration, and thus, the discrepancy between percentage of motile cells in HP and LP matrices (11.5% vs. 2%) is due to a six-fold increase in the ability of cells to form pore-like ‘channels’ in the matrix. Then, once the pore has been formed, migration occurs with identical mechanisms in both HP and LP matrices, leading to identical values for average speed, maximum speed, and displacement. The discussion attempts to summarize the point: “Under this explanation, the cells do not migrate until a channel of a sufficient width and distance is formed, and then, once such a channel is formed, migrate the length of that channel.”

What remains unclear is a) how long is the distance of the pore, and b) what happens after these pores have been formed? Is the cell limited by the distance of the pore that is formed? In Fig. 4h, the pore only appears to be 2-4 μm long.

We thank the reviewer for emphasizing the need for more clarity here. We have added a more succinct interpretation as suggested by Reviewer #1, stating that these data may suggest that the means by which cells form channels (i.e. using invadopodia protrusions)

and the means by which cells migrate through the channels are mechanistically decoupled.

In order to address the question regarding how long the distance of the pore is, we measured the fluorescence intensity of the RFP-LifeAct signal, which indicates the length of the actin-rich protrusion, prior to migration. For a representative migrating cell, a protrusion first extended about 26 μm , and then the cell migrated almost to the end of this channel, with the end of the cell body extending about $\sim 20 \mu\text{m}$ into the initially formed channel after the migration event (Figure R12 / Supplementary Fig. 7a). We note that the panels in Fig. 2e, 2i, 3a, and 3d are consistent with the idea that invasive protrusions can form channels up to 20 μm or more in length. With regard to what happens after these pores are formed, the data below, as well as the panels in Fig. 2e and 2i, indicate that the cells are indeed limited by the distance of the pore that is formed. We have clarified this idea in the manuscript text.

Figure R12: Final protrusion extending before migration event

In addition, the cell that has traversed this pore, which took many hours to build in the first place, was able to deform the matrix to the right of the pore to a degree equal to that on the left side, where the cell was initially encapsulated and thus the stress would have likely been much smaller. Furthermore, no densification seems apparent on the right side of Fig. 4h, despite a far higher degree of deformation that would have been present in the pore region. The only change that this reviewer believes is necessary to clear up this confusion is a more detailed cartoon diagram in figure 5 which hypothesizes exactly how the cell goes from 'New Mode' panel 2 to 'New Mode' panel 3. An additional panel in between these two time points showing the intermediate step would be very helpful. Does the entire cell body flow from the left side to the right side, to the limit of the furthest point of the invadopodia, then expand isotropically until we reach the 'New Mode' panel 3? Or did the invadopodia plastically displace the ECM to such a high degree that there is already a void in the matrix prior to cell translocation? To summarize, this reviewer is not requesting any new experiments and believes that the manuscript is suitable for publication, but believes it is in the best interest of the authors to clarify their proposed

'new mode' in a way that is in agreement with the data in Extended Figure 4 and the images in Figure 4 g-h.

We thank the reviewer for this suggestion. We have added additional panels into Fig. 5 (Figure R13), and comments into the figure's caption in the manuscript, to be consistent with the following 3 observations: (1) the channel formation occurs in stages, as invadopodia protrusions extend and retract; (2) the protrusions grow in both length and width before cell migration occurs; and (3) the final protrusion extension occurs in coordination with displacement of the cell body, which occupies the space made by the last, very long, very wide protrusion.

Figure R13: Summary Schematic

Response to Reviewer #4

General Comments:

The authors have done a great job in addressing all of my previous concerns. They have added significant additional experiments and analyses to support their conclusions. I feel that this manuscript should be published in its current form.

We thank the reviewer for a second reading of the manuscript and are pleased that Reviewer #4 thinks the manuscript should be published.

Bibliography

- Bonakdar, N., Gerum, R., Kuhn, M., Spörrer, M., Lippert, A., Schneider, W., Aifantis, K. E. and Fabry, B. (2016), 'Mechanical plasticity of cells', *Nature Materials*, **15**(10): 1090–1094.
- Caliari, S. R. and Burdick, J. A. (2016), 'A practical guide to hydrogels for cell culture', *Nature Methods*. Nature Publishing Group, **13**(5): 405–414.
- Carey, S. P., Rahman, A., Kraning-Rush, C. M., Romero, B., Somasegar, S., Torre, O. M., Williams, R. M. and Reinhart-King, C. A. (2015), 'Comparative mechanisms of cancer cell migration through 3D matrix and physiological microtracks', *American Journal of Physiology - Cell Physiology*, **308**(6): C436–C447.
- Chaudhuri, O., Koshy, S. T., Branco da Cunha, C., Shin, J.-W., Verbeke, C. S., Allison, K. H. and Mooney, D. J. (2014), 'Extracellular matrix stiffness and composition jointly regulate the induction of malignant phenotypes in mammary epithelium.', *Nature materials*, **13**(June): 1–9.
- Chen, M. B., Lamar, J. M., Li, R., Hynes, R. O. and Kamm, R. D. (2016), 'Elucidation of the Roles of Tumor Integrin $\beta 1$ in the Extravasation Stage of the Metastasis Cascade', 2513–2525.
- Denisin, A. K. and Pruitt, B. L. (2016), 'Tuning the Range of Polyacrylamide Gel Stiffness for Mechanobiology Applications', *ACS Applied Materials and Interfaces*, **8**(34): 21893–21902.
- Findley, W. N., Lai, J. S. and Onaran, K. (1976), *Creep and Relaxation of Nonlinear Viscoelastic Materials with an Introduction to Linear Viscoelasticity*, Dover Publications. New York: Dover Publications, Inc.
- Fisher, K. E., Pop, A., Koh, W., Anthis, N. J., Saunders, W. B. and Davis, G. E. (2006), 'Tumor cell invasion of collagen matrices requires coordinate lipid agonist-induced G-protein and membrane-type matrix metalloproteinase-1-dependent signaling', *Molecular Cancer*, **5**: 1–23.
- Franck, C., Maskarinec, S. a, Tirrell, D. a and Ravichandran, G. (2011), 'Three-dimensional traction force microscopy: a new tool for quantifying cell-matrix interactions.', *PloS one*, **6**(3): e17833.
- Friedl, P. and Wolf, K. (2010), 'Plasticity of cell migration: A multiscale tuning model', *Journal of Cell Biology*, **188**(1): 11–19.
- Hotary, K., Li, X., Allen, E., Stevens, S. L. and Weiss, S. J. (2006), 'A cancer cell metalloprotease triad regulates the basement membrane transmigration program', 2673–2686.
- Joubert, L.-M. (2017), 'Variable Pressure-SEM: a versatile tool for visualization of hydrated and non-conductive specimens', (February): 655–662.
- Kleinman, H. K. and Martin, G. R. (2005), 'Matrigel: basement membrane matrix with biological activity.', *Seminars in cancer biology*, **15**(5): 378–86.
- Kruegel, J. and Miosge, N. (2010), 'Basement membrane components are key players in specialized extracellular matrices', *Cellular and Molecular Life Sciences*, **67**(17): 2879–2895.
- Legant, W., Miller, J. and Blakely, B. (2010), 'Measurement of mechanical tractions exerted by cells in three-dimensional matrices', *Nature Methods*, **7**(12): 969–973.
- Marinkovich, M. P. (2007), 'Tumour microenvironment: laminin 332 in squamous-cell

- carcinoma.’, *Nature reviews. Cancer*, **7**(5): 370–80.
- Miller, D. R. and Peppas, N. A. (1986), ‘Bulk characterization and scanning electron microscopy of hydrogels of P(VA-co-NVP)’ , *Biomaterials*, **7**(5): 329–339.
- Moshfegh, Y., Bravo-Cordero, J. J., Miskolci, V., Condeelis, J. and Hodgson, L. (2014), ‘A Trio-Rac1-Pak1 signalling axis drives invadopodia disassembly.’, *Nature cell biology*, **16**(6): 571–585.
- Moss, N. M., Wu, Y. I., Liu, Y., Munshi, H. G. and Stack, M. S. (2009), ‘Modulation of the membrane type 1 matrix metalloproteinase cytoplasmic tail enhances tumor cell invasion and proliferation in three-dimensional collagen matrices’ , *Journal of Biological Chemistry*, **284**(30): 19791–19799.
- Nam, S., Hu, K. H., Butte, M. J. and Chaudhuri, O. (2016), ‘Strain-enhanced stress relaxation impacts nonlinear elasticity in collagen gels’ , *Proceedings of the National Academy of Sciences*, **113**(20): 201523906.
- Nam, S., Lee, J., Brownfield, D. G. and Chaudhuri, O. (2016), ‘Viscoplasticity Enables Mechanical Remodeling of Matrix by Cells’ , *Biophysical Journal*. Biophysical Society, **111**(10): 2296–2308.
- Poincloux, R., Collin, O., Lizárraga, F., Romao, M., Debray, M., Piel, M. and Chavrier, P. (2011), ‘Contractility of the cell rear drives invasion of breast tumor cells in 3D Matrigel.’ , *Proceedings of the National Academy of Sciences of the United States of America*, **108**(5): 1943–8.
- Prass, M., Jacobson, K., Mogilner, A. and Radmacher, M. (2006), ‘Direct measurement of the lamellipodial protrusive force in a migrating cell’ , *Journal of Cell Biology*, **174**(6): 767–772.
- Rowe, R. G. and Weiss, S. J. (2009), ‘Navigating ECM barriers at the invasive front: the cancer cell-stroma interface.’ , *Annual review of cell and developmental biology*, **25**: 567–595.
- Sabeh, F., Shimizu-Hirota, R. and Weiss, S. J. (2009), ‘Protease-dependent versus -independent cancer cell invasion programs: three-dimensional amoeboid movement revisited.’ , *The Journal of Cell Biology*, **185**(1): 11–9.
- Shames, I. H. and Cozzarelli, F. A. (1997), *Elastic and Inelastic Stress Analysis*. Washington, D.C.: Taylor & Francis.
- Sodek, K. L., Brown, T. J. and Ringuette, M. J. (2008), ‘Collagen I but not Matrigel matrices provide an MMP-dependent barrier to ovarian cancer cell penetration’ , *BMC Cancer*, **8**: 1–11.
- Wang, S. J., Saadi, W., Lin, F., Minh-Canh Nguyen, C. and Li Jeon, N. (2004), ‘Differential effects of EGF gradient profiles on MDA-MB-231 breast cancer cell chemotaxis’ , *Experimental Cell Research*, **300**(1): 180–189.
- Ward, I. M. and Hadley, D. W. (2004), *Introduction to the Mechanical Properties of Solid Polymers*. West Sussex, England: John Wiley & Sons Ltd.
- Wolf, K., Te Lindert, M., Krause, M., Alexander, S., Te Riet, J., Willis, A. L., *et al.* (2013), ‘Physical limits of cell migration: control by ECM space and nuclear deformation and tuning by proteolysis and traction force.’ , *The Journal of cell biology*, **201**(7): 1069–84.
- Yurchenco, P. D., Lu, P., Takai, K., Weaver, V. M., Hynes, R. O., Naba, A., *et al.* (2011), ‘Basement Membranes : Cell Scaffoldings and Signaling Platforms’ , *Cold Spring Harbor perspectives in biology*, **3**: 1–28.

- Zervantonakis, I. K., Hughes-Alford, S. K., Charest, J. L., Condeelis, J. S., Gertler, F. B. and Kamm, R. D. (2012), 'Three-dimensional microfluidic model for tumor cell intravasation and endothelial barrier function', *Proceedings of the National Academy of Sciences*, **109**(34): 13515–13520.
- Zhao, X., Huebsch, N., Mooney, D. J. and Suo, Z. (2010), 'Stress-relaxation behavior in gels with ionic and covalent crosslinks.', *Journal of applied physics*, **107**(6): 63509.

RESPONSE TO REVIEWERS

Response to Reviewer #1

Wisdom et al. presented a much-improved version of their manuscript. The authors addressed most of my previous concerns about the paper. The data on the plasticity measurements of different synthetic and biologically relevant ECMs are very much appreciated. However, two remaining points should be addressed:

We thank Reviewer #1 for the careful review and supportive comments. We are glad to see that we have addressed most of this reviewer's previous concerns.

1) It is unfortunate that the authors did not conduct in vivo experiments to validate the in vivo relevancy of their proposed migration mode, although the reviewer acknowledges the difficulty in setting up an adequate mouse model. Lacking in vivo confirmation, the authors should clarify in the discussion that the mode of migration they observed in vitro has not yet been observed in vivo. Similarly, the wording on the last paragraph of the discussion, in which the authors discuss the clinical relevancy of their finding, should be toned down since no in vivo (murine or other) experimental data were provided to support the in vivo relevancy of their findings.

We have clarified that the mode of migration observed *in vitro* has not been observed *in vivo* in the discussion section of the manuscript and toned down the wording for the last paragraph of the discussion.

2) Regarding minor comment #5 from the last round of the review, the authors' argument that the comparison of the degree of force generation by 4T1 and MDA231 tumor cells is beyond the scope of this manuscript was not compelling. Since the authors are claiming a novel form of migration that depends solely on the ability of cells to generate force, a careful characterization and comparison between different cell types' ability to generate force and how it relates to cell migration in the synthetic matrix is of central importance. This is especially true since 4T1 tumors can form invadopodia and are known to be highly invasive and aggressive in the murine model. While not required, this reviewer would recommend that the authors consider performing a simple experiment to compare the traction forces generated by the 4T1 and MDA231 tumor cells. This would be a useful addition to the paper.

We thank the reviewer for the suggestion and agree that these results would be interesting. However, due to differences between 4T1s and MDA-MB-231s in terms of genetics, morphology, and migration characteristics, we would not be able to draw any definitive conclusions regarding the relationship between force generation and migration from this simple experiment. A more definitive understanding of the relationship between force generation (both degree and timescale) and migration in the synthetic matrices shown here would require greater spatiotemporal resolution than our experiments can currently accommodate, and the analysis of many different cancer cell lines to control for genetic differences. Since the reviewer has noted that this is not required, we have opted to leave this experiment out of the manuscript.

Response to Reviewer #2

The authors have done a careful job of addressing all of the comments once again and they have addressed all of my concerns adequately. Now that the authors provide more convincing evidence that their alginate gels are plastic and deform in a permanent way, it emphasises the novel aspects of the work. I very much like the model in Figure R13 - it helps to explain the concepts here to biologists and it highlights the novelty of the observations.

We thank Reviewer #2 for the careful review of the manuscript and their support for publication.